# Meta-analysis reveals weak but pervasive plasticity in insect thermal limits

Hester Weaving [1] ✉, John S. Terblanche [2,4], Patrice Pottier [3] &
Sinead English [1,4]

Extreme temperature events are increasing in frequency and intensity due to climate change. Such events threaten insects, including pollinators, pests and disease vectors. Insect critical thermal limits can be enhanced through acclimation, yet evidence that plasticity aids survival at extreme temperatures is limited. Here, using meta-analyses across 1374 effect sizes, 74 studies and 102 species, we show that thermal limit plasticity is pervasive but generally weak: per 1 °C rise in acclimation temperature, critical thermal maximum increases by 0.09 °C; and per 1 °C decline, critical thermal minimum decreases by 0.15 °C. Moreover, small but significant publication bias suggests that the magnitude of plasticity is marginally overestimated. We find juvenile insects are more plastic than adults, highlighting that physiological responses of insects vary through ontogeny. Overall, we show critical thermal limit plasticity is likely of limited benefit to insects during extreme climatic events, yet we need more studies in under-represented taxa and geographic regions.

Extreme heatwaves are becoming more frequent and intense, whilst the reverse is true for extreme cold events[1]. The upper and lower critical thermal limits (CTLs) of animals, frequently estimated as critical thermal maximum and minimum ($CT_{max}$, $CT_{min}$ respectively), serve as useful proxies for inferring climate-related vulnerability[2,3]. Extreme heatwaves are expected to exceed species' CTLs, so animals must adapt or move poleward to cooler climes[4]. As high latitudes have greater variation in surface temperature, both poleward advancement and maintenance of current ranges will expose species to a greater frequency and magnitude of extreme temperatures[5,6]. Plasticity of CTLs—a flexible response to changing conditions that can occur at the level of individuals, populations, or species—provides an important mechanism for populations to enhance tolerance, and cope with increasingly variable and intense temperatures[2]. Such plasticity can be achieved through acclimation, whereby prior thermal exposure can cause a shift in CTLs, allowing animals to perform better at, or recover from, more extreme temperatures[7,8]. For example, acclimation can cause upregulation of heat shock proteins, and results in changes to the phospholipid composition of the cell membrane[9,10]. Plasticity could therefore be important for tracking increasingly variable and intense temperatures and allow time for evolutionary responses via slower genetic change across generations[11].

Insects fulfil diverse ecological roles as pollinators, agricultural pests and disease vectors, and there is global concern over recent, rapid declines in abundance of rare, ecologically- or agriculturally-important species and, conversely, spikes in pest outbreaks[12,13]. How insects will respond to climate change via plasticity remains an important topic of debate[14,15]. Recent systematic reviews and formal meta-analyses across ectotherms have assessed plasticity of CTLs and described broad-scale patterns of variation in plasticity[16–21]. Generally, these studies find weak plasticity of CTLs, concluding that this mechanism has limited potential to aid survival of ectothermic species under climate change. Explaining broad-scale trends is, however, complicated, and contradictory findings have been presented regarding the relationship of plasticity with factors such as latitude, seasonality, and body size[16,17,22]. With the general focus on ectotherms, trends specific to important assemblages—such as insects—may be obscured, and traits unique to these assemblages, for example development type, are typically not investigated.

[1]School of Biological Sciences, University of Bristol, Bristol, UK. [2]Department of Conservation Ecology & Entomology, Stellenbosch University, Stellenbosch, South Africa. [3]Ecology & Evolution Research Centre, School of Biological, Earth and Environmental Sciences, The University of New South Wales, Sydney, NSW, Australia. [4]These authors jointly supervised this work: John S. Terblanche, Sinead English. ✉ e-mail: hester.weaving@bristol.ac.uk

Here, we undertake a systematic meta-analysis of experimental studies on the plasticity of insects' upper and lower critical thermal limits, including taxon-specific moderators to investigate variation in plasticity. We investigate plastic responses to thermal acclimation – measured as the acclimation response ratio (ARR), the change in CTL with a given change in acclimation temperature– as they are relevant to future climate change scenarios, widely reported, and have been used in previous meta-analyses on the topic[16–20]. We examined (a) insects' ability to adjust their CTLs via plasticity, (b) broad-scale trends across origin (latitude and habitat type), ecology and morphology (sex and body size), and ontogeny (life stage and development type), and (c) how diverse methodologies used in experimental studies affect plasticity estimates. We outline our key predictions at the beginning of the Results section.

Overall, we show that critical thermal limits have generally weak plasticity, in keeping with the broader literature. Evidence of publication bias, although of small effect, indicates that insects could be even less plastic than previously predicted. Few broad-scale trends were identified, suggesting that insects express complex and heterogenous responses to their thermal environment. We also found that juvenile insects were more plastic than adults, indicating that insects can better compensate for variable temperatures during development.

## Results
### A priori predictions
A priori predictions for meta-analysis moderators were used to examine variation in plasticity of critical thermal limits (CTLs) in insects. We make general predictions but note that often contradicatory arguments can be made and responses may be mediated by other factors (e.g., trade-offs with basal resistance, the temperature-size rule, mobility, and organismal biochemical and physiological constraints)[19,23–25].

1. Origin
Theory suggests that selection drives plastic responses in animals from environments with moderate environmental variability and a degree of predictability[26,27].

   (a) Latitude
   We expect animals from lower latitudes to show less thermal tolerance plasticity than those living at higher latitudes, which have higher seasonality and thus predictable variability[28,29]. We acknowledge, however, that these predictions have received mixed support both in quantitative syntheses on ectotherms[16,17,19,22], and when explicitly tested in insects at either the species[30] or population[31] level.

   (b) Habitat type
   We predict that terrestrial organisms will have greater thermal tolerance plasticity than aquatic species, where the environment is more stable. However, here too some evidence suggests the contrary[16].

2. Ecology and morphology
   (a) Size
   Larger insects tend to be longer lived than smaller insects so are likely exposed to a greater range of temperatures throughout their lifetimes[17]. We therefore predict that ARR will increase with body size. Evidence consistent with this prediction has been found in quantitative syntheses on ectotherms[17]. However, first principles also suggest that plasticity of thermal tolerance could decrease with size, due to greater thermal inertia in smaller insects, and reduced ability to exploit microclimates[32,33].

   (b) Sex
   Given that animals are often sexually dimorphic, CTL plasticity may differ between sexes. For example, across ectothermic animals, males are often smaller than females[34]. However, males also tend to display more risky behaviours which could expose them to greater temperature variability, promoting selection for

thermal tolerance plasticity[34,35]. Due to conflicting selection pressures, we predict no consistent difference between sexes in insects. When this hypothesis was tested across ectothermic animals by meta-analysis, either no differences were found, or females had greater plasticity of thermal tolerance[36].

3. Ontogeny
Animals express different degrees of plasticity within their lifetimes[37–39]. Insect life stages often differ considerably in traits such as size and behaviour, often utilising distinct niches. As an added complication, insects under high developmental temperatures generally become smaller, following the temperature-size rule (TSR)[32], which may act counter to our predictions. A formal test of TSR could not be undertaken in the present study as most mass estimates were derived from the wider literature (see Methods for specific details).

   (a) Life stage at acclimation
   We expect that juvenile insects will have greater CTL plasticity early in life, due to juvenile stages being less mobile than adults and so less able to regulate temperature behaviourally[39].

   (b) Development type
   We predict that hemimetabolous insects will have greater plasticity than holometabolous insects. Developmental plasticity in holometabolous insects may be lost after metamorphosis due to morphological reorganisation[37]. As hemimetabolous insects undergo gradual development, rather than complete metamorphosis, plasticity may be preserved into adulthood.

4. Methodology
Methodology regarding how CTLs should be measured has been widely debated[40]. Less frequently acknowledged is how these diverse measures affect plasticity estimates[41,42]. Typically, studies use dynamic assays, where, following a period of acclimation, temperature is ramped until a predefined endpoint ($CT_{max}$ or $CT_{min}$). However, specific methodology necessarily varies widely to accommodate diverse taxa and life stages with unique behaviours, which in turn may affect comparability across studies and species[43].

   (a) Duration of acclimation treatment
   The duration of acclimation treatment can vary from hours to weeks. As acclimation can be stressful, we expect smaller ARRs for longer acclimation times as injury accumulates exponentially with time (e.g., discussed in[44,45]). However, we acknowledge that under mildly stressful conditions, more time under acclimation can allow for increased plasticity[46].

   (b) Assay ramp rate
   The ramp rate can vary by a factor of 75, with some studies arguing that faster rates have greater ecological relevance[47]. We predict that a slower ramp rate will generally result in reduced CTL plasticity due to increased time for measurable divergence between control and treatment groups. However, we acknowledge that the effect of ramp rate on CTL plasticity has not been well explored in the insect literature (but see ref. 48 for evidence in springtails).

   (c) Endpoint definition
   A temperature ramp causes a series of behavioural and physiological responses in insects, such as loss of coordination, partial paralysis, muscle spasms, and finally, total paralysis and death. We expect greater plasticity in CTLs at behavioural endpoints taken further from lethal conditions (i.e., more benign) than if the endpoint is measured as, or at, death[49].

   (d) Insect source
   Multiple environmental factors vary in the field (e.g., temperature and photoperiod) which may magnify plastic responses, in contrast to controlled conditions of the laboratory environment where plasticity may be less pronounced, especially if acclimation response is stimulated by a single factor at a time (e.g.,

temperature alone). Therefore, we predict that studies which use laboratory populations will find less pronounced plasticity of CTLs than those which use field-caught insects[50].

## Effect size dataset

A total of 803 effect sizes (from 60 studies, 92 species) were analysed for $CT_{max}$ and 571 (from 52 studies, 74 species) for $CT_{min}$. Overall, the analysis for both measures comprised 102 species from 74 studies. Diptera were by far the most represented order (k = 684; with most effect sizes from Drosophilidae (k = 584)), followed by Coleoptera (k = 261), Hemiptera (k = 150), Hymenoptera (k = 101), Lepidoptera (k = 75), Blattodea (k = 26), Trichoptera (k = 26), Ephemeroptera (k = 20), Plecoptera (k = 17), Odonata (k = 6), Grylloblatta (k = 6) and Orthoptera (k = 4).

## Overall plasticity of critical thermal limits

Overall, we found weak, positive plasticity for both upper and lower CTLs. For every 1 °C rise in acclimation temperature, $CT_{max}$ increased by 0.091 °C (95% CI = 0.030, 0.153; Fig. 1). Lower thermal limits were 60% more plastic; $CT_{min}$ decreased by 0.147 °C (95% CI = 0.106, 0.188; Fig. 1) for every 1 °C decline in acclimation temperature. Full statistics can be found in Supplementary Table 2.

## Broad-scale patterns in critical thermal limit plasticity

We assessed whether variation in plasticity of CTLs was explained by moderators using a series of univariate (Supplementary Tables 3-4) and multivariate models (Supplementary Tables 5–10). Due to 15–35% of data missing for latitude, acclimation duration and mass, we did not include these moderators in the multivariate models. ARRs are stated as mean differences between groups (with the direction of comparison stated in subscript) or as meta-regressions.

We expected insects originating from environments with greater temperature variability to be more plastic, however we found no relationship between latitude and ARR ($CT_{max}$ βARR = −0.001; 95% CI = −0.002, 0.001; $CT_{min}$ βARR = −0.001; 95% CI = −0.002, 0.001) and no difference between aquatic and terrestrial insects ($CT_{max}$ ARR $_{terrestrial-aquatic}$ = 0.002; 95% CI = −0.111, 0.117; $CT_{min}$ ARR $_{terrestrial-aquatic}$ = 0.115; 95% CI = −0.067, 0.297). We predicted that insects with larger mass would have greater ARRs, however we did not find any relationship ($CT_{max}$ βARR = 0.001; 95% CI = −0.001, 0.003; $CT_{min}$ βARR < −0.001; 95% CI = −0.001, < 0.001). As expected, we found no overall difference in thermal tolerance between male and female insects ($CT_{max}$ ARR $_{male-female}$ = 0.035; 95% CI = −0.005, 0.076; $CT_{min}$ ARR $_{male-female}$ = −0.028; 95% CI = −0.098, 0.042). For full comparisons and individual coefficients, see Supplementary Tables 3 and 4.

We predicted that plasticity of CTLs would be greater in juveniles and that holometabolous insects would be less plastic than hemimetabolous insects. As anticipated, we found that insects acclimated in adulthood were less plastic than insects acclimated during early life (Fig. 2; $CT_{max}$ ARR $_{adult-early\ life}$ = −0.036; 95% CI = −0.066, −0.007; $CT_{min}$ ARR $_{adult-early\ life}$ = −0.067; 95% CI = −0.131, −0.003), although only 1.3% and 1.6% of the variation was explained for $CT_{max}$ and $CT_{min}$ respectively. We also found that upper thermal limits of holometabolous insects were less plastic than those of hemimetabolous insects, explaining 5.8% of variation (Fig. 3; $CT_{max}$ ARR $_{holo-hemi}$ = −0.090; 95% CI = −0.175, −0.006). However, when Orthoptera (k = 1) was excluded from the analysis, this result was no longer significant ($CT_{max}$ ARR $_{holo-hemi}$ = −0.064; 95% CI = −0.131, 0.004). For lower thermal limits, we found no significant difference in plasticity between the two development types, although the trend was also for lower plasticity in holometabolous insects (Fig. 3; $CT_{min}$ ARR $_{holo-hemi}$ = −0.036; 95% CI = −0.132, 0.059). To investigate differences between development types before metamorphosis, juvenile insects were analysed as a subset. Again, we found there was no significant difference in CTL plasticity between hemi- and holometabolous insects at the

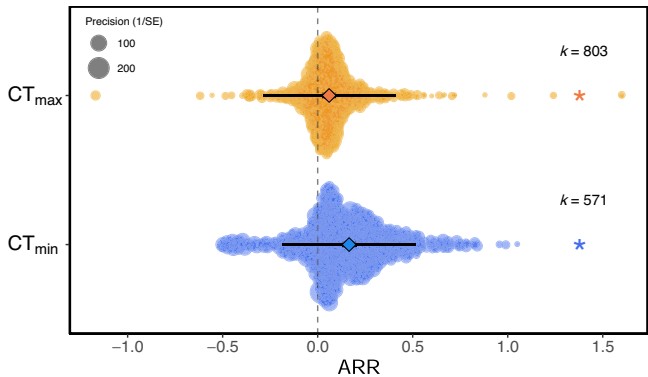

**Fig. 1 | Meta-analytic mean acclimation response ratio (ARR) for upper and lower critical thermal limits, $CT_{max}$ (k = 803) and $CT_{min}$ (k = 571).** A positive ARR indicates an adaptive plastic response; heat acclimation increases $CT_{max}$ or cold acclimation decreases $CT_{min}$. 95% confidence intervals (95% CIs) are depicted in heavy black lines (and partially hidden by the mean data points, depicted by a diamond symbol), prediction intervals in thin black lines. The size of each data point is proportional to the precision of the study (1/SE (Standard Error)). k = number of effect sizes per group. Asterisk indicates that 95% CIs do not span zero.

juvenile stage, although, again, there was a trend for lower plasticity in holometabolous insects ($CT_{max}$ ARR $_{holo-hemi}$ = −0.057; 95% CI = −0.137, 0.024; $CT_{min}$ ARR $_{holo-hemi}$ = −0.063; 95% CI = −0.183, 0.057).

Plasticity in CTLs varied depending on the assay endpoint employed. If $CT_{max}$ was defined as when the insect lost its righting response, the plasticity was significantly greater than all other endpoints, excluding death (Fig. 4; for full comparisons see Supplementary Table 3). For $CT_{min}$, when the response was measured as death, CTLs were less plastic than when the endpoint was measured as loss of clinging, righting, or motor control. When the endpoint was measured as loss of natural position, ARRs were lower than all other endpoints, excluding death and loss of activity (Fig. 4; for full comparisons see Supplementary Table 4). We expected longer times under stressful conditions to result in smaller ARRs, however we found no relationship with acclimation duration ($CT_{max}$ βARR < −0.001; 95% CI < −0.001, < 0.001; $CT_{min}$ βARR < 0.001; 95% CI < −0.001, < 0.001), or assay ramp rate ($CT_{max}$ βARR = 0.020; 95% CI = −0.067, 0.106; $CT_{min}$ βARR = 0.017; 95% CI = −0.091, 0.125). We predicted that field-caught insects would be more plastic than laboratory-reared insects. However, we found no difference for upper thermal limits ($CT_{max}$ ARR $_{lab-field}$ = 0.021; 95% CI = −0.044, 0.086) and opposing evidence for lower thermal limits, where laboratory insects were more plastic than field-caught insects and 1.2% of variation was explained ($CT_{min}$ ARR $_{lab-field}$ = 0.052; 95% CI = 0.006, 0.098).

Multivariate models were used to find the best fitting models ranked by AICc. For upper thermal limits, our best model included development type as the only moderator, finding holometabolous insects were less plastic than hemimetabolous insects (Supplementary Table 5). Our second-best model also indicated that holometabolous insects were less plastic, and found insects acclimated in early life had greater ARRs than those acclimated during adulthood, consistent with our univariate models (Supplementary Table 5). These models explained 5.9% and 6.4% of variation respectively. Our model averaging approach using conditional averages showed that the most important moderators ranked by AICc were development type, life stage at acclimation, source, and ramp rate (Supplementary Table 7; see Supplementary Table 8 for full averages). Development type and life stage at acclimation were significant moderators. Differences between $CT_{max}$ endpoint methodologies were not robust to model averaging, indicating that differences are likely driven by other moderators. For lower thermal limits, the best model indicated that

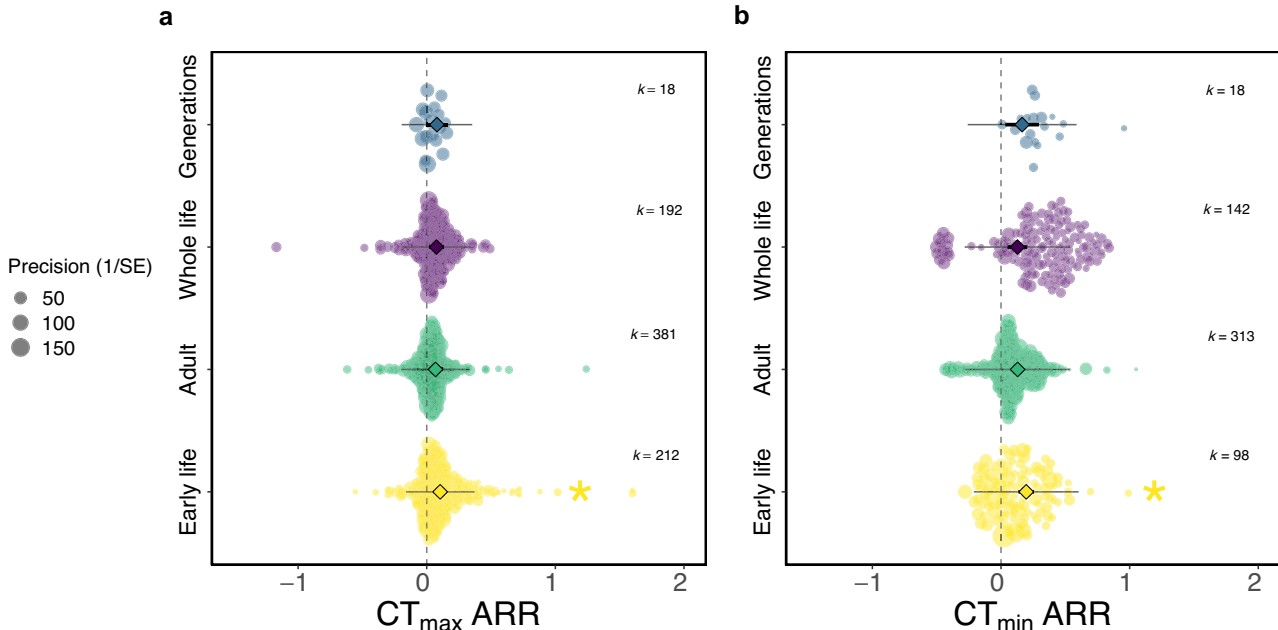

**Fig. 2 | The effect of acclimation life stage on acclimation response ratio (ARR) for critical thermal limits, a CT$_{max}$ (k = 803) and b CT$_{min}$ (k = 571).** Early life is defined as the egg or nymph stage for hemimetabolous (non-metamorphosising) insects and the egg, larval or pupal stage for holometabolous (metamorphosising) insects. 95% confidence intervals (95% CIs) are depicted in heavy black lines (often hidden by the mean data point, depicted by a diamond symbol), prediction intervals in thin black lines. The size of each data point is proportional to the precision of the study (1/SE (Standard Error)). k = number of effect sizes per group. Asterisk indicates that 95% CIs do not overlap, comparisons are made with the adult group.

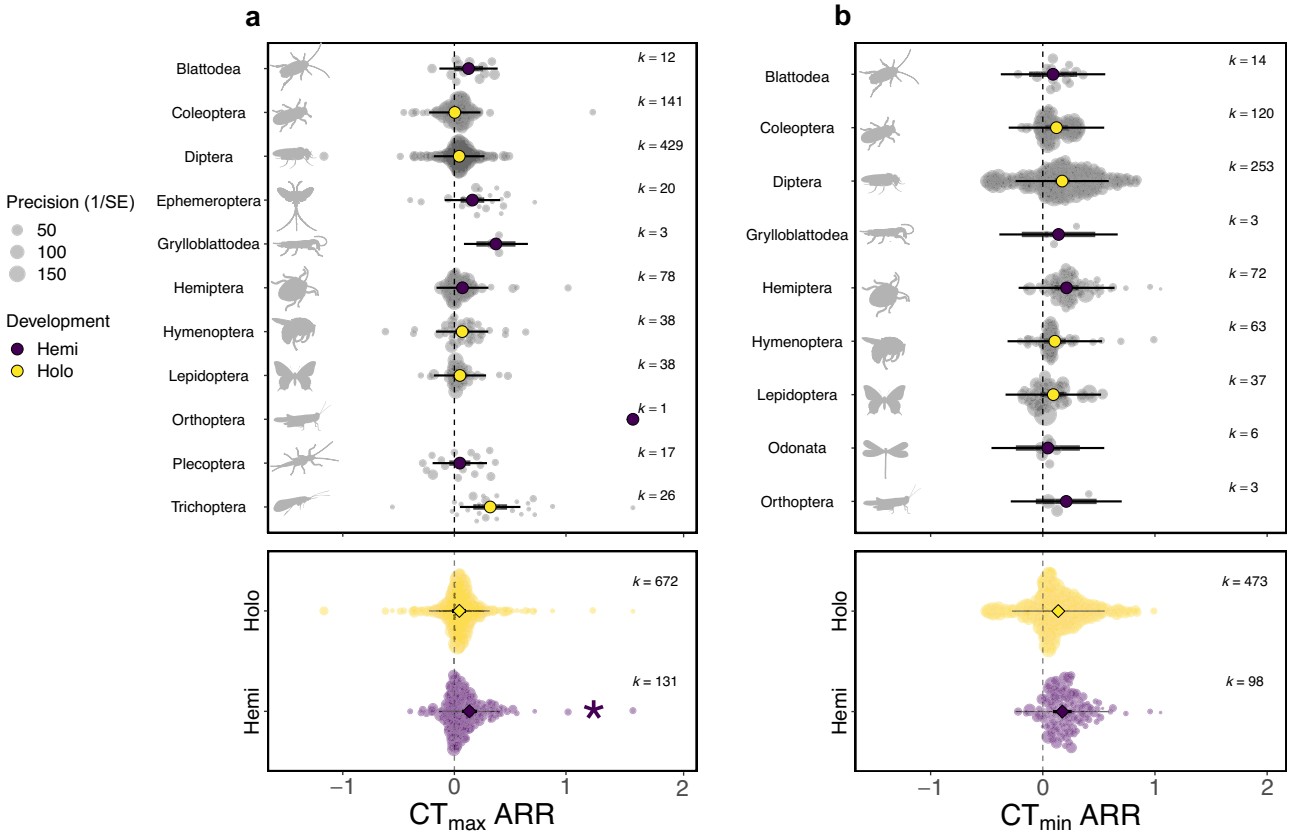

**Fig. 3 | The effect of development type (holometabolous, metamorphosing insects; or hemimetabolous, non-metamorphosing insects) on acclimation response ratio (ARR) for a CT$_{max}$ (k = 803) and b CT$_{min}$ (k = 571).** Mean ARR are arranged by Insect Order (alphabetical), coloured by development type. 95% confidence intervals (95% CIs) are depicted in heavy black lines (sometimes hidden by the mean data point, depicted by a diamond or circle symbol), prediction intervals in thin black lines. The size of each data point is proportional to the precision of the study (1/SE (Standard Error)). k = number of effect sizes per group. Asterisk indicates that 95% CIs do not overlap, comparisons are made between the two development types. Icon credit: phylopics.

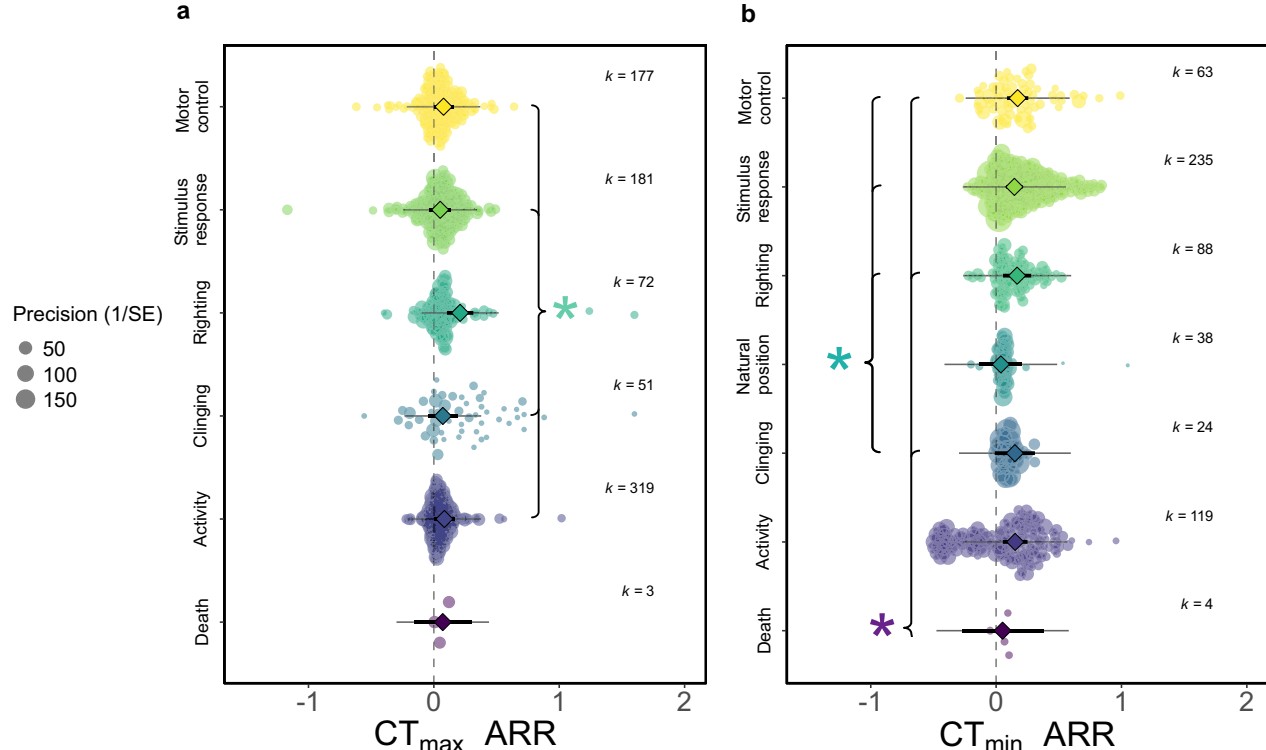

**Fig. 4 | Difference in acclimation response ratio (ARR) between assay endpoint definitions of critical thermal limits, a CT$_{max}$ ($k$ = 803) and b CT$_{min}$ ($k$ = 571).** The endpoint of the assay is the behaviour at which the critical thermal limit is taken. Excluding 'Death', variables can be read as 'Loss of…'. 95% confidence intervals (95% CIs) are depicted in heavy black lines (sometimes hidden by the mean data point, depicted by a diamond symbol), prediction intervals in thin black lines. The size of each data point is proportional to the precision of the study (1/SE (Standard Error)). $k$ = number of effect sizes per group. Asterisk indicates that 95% CIs do not overlap, brackets on the right show that the reference group is larger than marked groups, on the left, smaller.

laboratory insects are more plastic than field-caught insects, consistent with our univariate model (Supplementary Table 6). The most important moderators using conditional averages were source, life stage at acclimation, sex, habitat, development type and ramp rate (Supplementary Table 9; see Supplementary Table 10 for full averages). Model averaging indicated that juvenile insects had greater plasticity in lower thermal limits than adult insects. This approach also indicated that insects from the unknown sex group were less plastic than female insects.

**Heterogeneity, publication bias and sensitivity analysis**
Heterogeneity was very high (CT$_{max}$ I$^2$ = 97%; CT$_{min}$ I$^2$ = 99%, for intercept models), as common in ecological and evolutionary meta-analyses[51]. Random factors explained heterogeneity; for upper thermal limit ARR, differences between studies explained 15.1%, phylogeny explained 15.3%, non-phylogenetic differences between species explained 17.7%, and effect size ID explained 49.1%. For CT$_{min}$ ARR, phylogenetic and non-phylogenetic signals were far weaker, both explaining <0.1% of the variation. Otherwise, study ID and effect size ID explained 36.1% and 63.0% of heterogeneity respectively.

The leave-one-out sensitivity analysis showed that no species, family or study had a disproportionate impact on results (Supplementary Tables 11 and 12). Sensitivity analysis excluding Drosophilidae and studies with fluctuating temperatures during acclimation also showed no disproportionate impact of these studies (Supplementary Tables 13 and 14).

Funnel plots for plasticity in CT$_{max}$ and CT$_{min}$ are shown in Fig. 5. Egger's regression test revealed significant publication bias for CT$_{max}$ ARR intercept model (Supplementary Fig. 3; CT$_{max}$ βARR = 0.288; 95%

CI = 0.028, 0.548) and best model (CT$_{max}$ βARR = 0.288; 95% CI = 0.028, 0.548). The mean ARR corrected for publication bias was 0.0907 °C (rather than 0.0913 without the correction) (Supplementary Table 19; 95% CI = 0.030, 0.152). We also found significant publication bias for CT$_{min}$ ARR, with the model predicting 0.144 °C per degree change, rather than 0.147 °C (Supplementary Table 22; 95% CI = 0.102, 0.185). This result was found for both the intercept model (Supplementary Fig. 3; CT$_{min}$ βARR = 0.621; 95% CI = 0.068, 1.174), and the best model (CT$_{min}$ βARR = 0.695; 95% CI = 0.140, 1.249). For full model outputs see Supplementary Tables 18-23. We found no evidence of a relationship between publication year and ARR for either CT$_{max}$ (Supplementary Table 24; CT$_{max}$ βARR = −0.001; 95% CI = −0.005, 0.003) or CT$_{min}$ (Supplementary Table 25; CT$_{min}$ βARR = −0.002; 95% CI = −0.009, 0.005).

## Discussion
We found that both upper and lower critical thermal limits of insects had weak but pervasive plasticity, with a mean shift of 0.092 °C and 0.147 °C respectively in response to a 1 °C adjustment in acclimation temperature. Evidence for small but significant publication bias suggests that responses are likely to be a fraction more modest than reported here and in the wider literature (where such bias has not been previously investigated). These findings are in agreement with broader comparisons across ectotherms, showing thermal limit plasticity is generally weak[16–18,20]. Indeed, in Gunderson and Stillman's 2015[16] analysis, insects had the weakest responses of all ectothermic groups and Morley et al.'s 2019[20] analysis illustrated a similar pattern (when excluding high latitude species). Under our current climate, some evidence suggests that the majority of ectothermic species are close to, or without, a thermal safety margin, as operative body

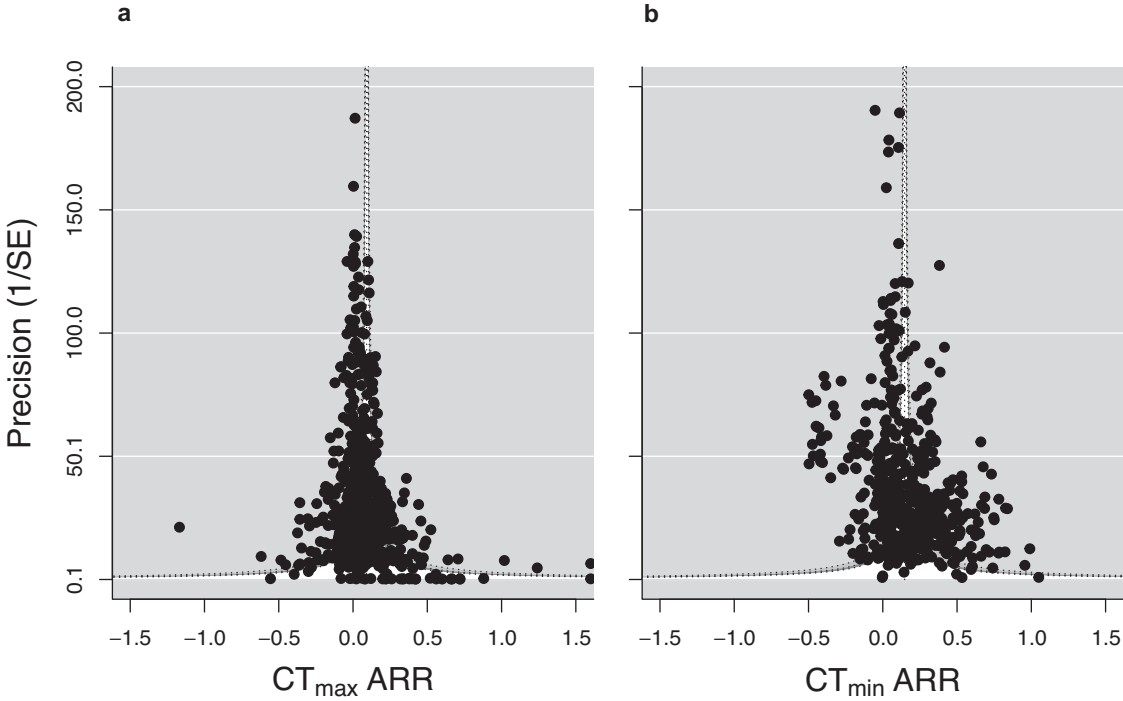

**Fig. 5 | Publication bias in acclimation response ratios (ARR) of critical thermal limits, a $CT_{max}$ (k = 803) and b $CT_{min}$ (k = 571).** More precise studies (those with higher 1/SE (Standard Error) are located at the top of the plot, and less precise studies are located at the bottom. Egger's regression test (two-sided) showed slight significant positive publication bias for $CT_{max}$ ($CT_{max}$ βARR = 0.288; 95% CI = 0.028, 0.548; $p = 0.030$) and $CT_{min}$ ($CT_{min}$ βARR = 0.621; 95% CI = 0.068, 1.174; $p = 0.028$) intercept models, indicating that data points are missing from the left-hand side of both plots. Shadings indicate ($p < 0.05$, $p < 0.01$, $p < 0.001$).

temperatures in exposed environments often match or exceed physiological limits (albeit requiring a number of simplifying assumptions)[3]. With once-in-a-decade heat wave events expected to be at least four times more likely, most ectotherms – and given the current evidence, especially insects – will need to rely on other compensatory mechanisms[1,3]. For example, insects can behaviourally thermoregulate using microclimates, e.g., leaf shade in a forest canopy can reduce maximum air temperature by 5 °C[23,52]. Poleward migration has also been documented in numerous insect species, and is favoured by their short generation times, fast reproduction and high mobility[5]. Generally weak plasticity of insect CTLs may be of some added benefit when working in combination with these mechanisms, particularly in species with range shifts into more variable poleward regions[5,6].

While most ecological and morphological moderators did not significantly explain variation in CTL plasticity, our study indicates a potentially important role of ontogeny. We found that insects acclimated in early life had greater plasticity than those acclimated in adulthood, providing support for our hypothesis that juveniles are more plastic than adults. This indicates the presence of a sensitive window, where acclimation elicits a greater response in early life stages. Variation in plasticity over an insect's lifetime reflects changes in the costs and benefits of plasticity. For example, plasticity early in ontogeny may have evolved due to juvenile insects being less able to behaviourally thermoregulate since they have generally lower motility than adults[37]. This may mean that juvenile insects are exposed to greater variation in temperature, thereby promoting selection for greater plasticity[26,27]. There also may be differences in the frequency and reliability of cues earlier in life, particularly apparent in insect life cycles where juvenile and adult stages utilise entirely different niches[39]. More generally, phenotypic adjustments earlier in life can have a greater impact on fitness, because fewer individuals survive or reproduce later on in life, thus selection tends to decrease with age[38]. Overall, these findings suggest that plasticity in juvenile insects may be critical to later thermal tolerances and suggest that developmental

effects should be further investigated for their relevance to insect climate change responses.

We also found some evidence that hemimetabolous insects have greater plasticity of upper thermal limits than holometabolous insects. Any developmental plasticity in holometabolous insects may be lost through metamorphosis, due to dramatic cell, tissue and whole-animal reorganisation, likely contributing to lower plasticity[37]. This may serve an adaptive function, as cues are less comparable across life stages in holometabolous insects, where juveniles are immobile in the pupal stage and larvae often have different ecologies to adults[39]. However, evidence for differences between developmental types was not robust to the exclusion of Orthoptera (k = 1), despite all four models (including subset data, see Supplementary Tables 3 and 4) returning results in the same direction. There is a clear need for more standardised critical thermal limit studies on Orthopterans, and hemimetabolous insects in general, to determine whether plastic differences in CTL between development types are robust or an artifact of low sample sizes.

We found variation in plasticity of critical thermal limits between some types of methodology. The definition of the endpoint used in any given study led to significant differences in plasticity for both upper and lower CTLs. However, differences were not found in model averages so are likely driven by other sources of variation. Additionally, contrary to expectations, we found evidence for greater plasticity of lower thermal limits in laboratory-reared insects than in field-caught insects. This could be due to more factors influencing and interacting with tolerance in field-collected individuals, while in the laboratory, more factors (e.g., diet, age, thermal history all affect estimates) are controlled. Consequently, the signal in laboratory studies is clearer: that is, more distinct from 'noise' or variation. Notably, no relationship was found between ARR and acclimation duration or ramp rate, perhaps owing to complex interactions which were not investigated in this analysis, such as those between ramp rate and acclimation, nutrition and body condition, and interval time between the acclimation

treatment and thermal assay endpoint[48,53]. Our preliminary analyses investigating the temperature-size rule (Supplementary Table 3 and 4) found no difference between groups where acclimation treatment and CTL assay were within a life stage, or over different life stages. However, it would be interesting to investigate this further where study-specific mass or size data are available. Our findings indicate the need to consider diverse aspects of methodology and population history in future comparative analyses.

Our study adds to evidence that upper thermal limits are less plastic and more evolutionarily constrained than lower thermal limits[54]. We found $CT_{max}$ was ~60% less plastic than $CT_{min}$ which may reflect the distinct physiological and biochemical responses at the two extremes of temperature. $CT_{max}$ is often lethal, with loss of function occurring at the same temperature as heat death. There are relatively few studies on the mechanisms of heat death, but the breakdown of membrane function may cause the impairment of ion pumps, nutrient transport and mitochondrial function, leading to the loss of metabolic control and homeostasis, and, finally, cell death[55]. In contrast, $CT_{min}$ usually results in a non-lethal chill coma, whereafter an insect may recover fully. The mechanisms of chill coma are also poorly understood, but are likely driven by the breakdown of ionic homeostasis due to the effect of low temperature on ATPases, ion channels and the lipid membrane[56]. We detected a phylogenetic signal in $CT_{max}$ in our models, which was not observed for $CT_{min}$. This may reflect evolutionary constraints for $CT_{max}$, such as high fitness costs or substantial genetic changes required to modify upper thermal limits, causing related species to share similar thermal responses[57]. If upper thermal limits cannot evolve easily due to these constraints, an organism's current thermal limits will dictate the kind of environments in which it can survive. These differences create a 'concrete ceiling' for $CT_{max}$, where physiological barriers prevent extensive evolution and perhaps also restrict plasticity[58]. As extreme heatwaves are becoming more frequent and intense, and extreme cold events less so, concrete ceilings will likely create strong barriers to adaptation for insect species. Understanding which species have hidden or multivariate adaptive capacity would then be essential to forecasting species responses to climate change.

Here, we focused on critical thermal limits because they are considered an important predictor for climate change and are well-studied. However, there is some evidence that CTLs do not correlate well with species abundance or distribution, and therefore might not be the best predictors for assessing climate change impacts[59,60] (but see[3]). In contrast, thermal fertility limits, the temperature at which an animal becomes sterile, can be far more sensitive to temperature, with evidence in ectotherms suggesting that these occur at less extreme temperatures[61,62]. Our study also highlights the need for greater taxonomic diversity in CTL measures, representative of broad biogeographic regions and development types, as nearly one third of the effect sizes in our study were from Drosophilidae species (k = 584 out of 1374 total; note that a comprehensive exploration of effect sizes *within* Drosophilidae species is presented in Supplementary Tables 15-17), and hemimetabolous insects were not well represented (k = 229).

We found that plasticity of insect critical thermal limits was positive but weak, supporting previous findings for ectotherms more broadly. Detection of a phylogenetic signal for upper, but not lower, thermal limits indicates that evolutionary adaptation may also be constrained for $CT_{max}$. Ontogenetic variation in CTL plasticity suggests that a developmental window may be important in shaping insects' responses to changes in temperature and these effects should be incorporated in climate vulnerability assessments. Overall, most insect species will need to rely extensively on distributional changes and behavioural regulation if they are to buffer the effects of climate change.

## Methods

### Literature search

Each step was reported according to the PRISMA (Preferred Reporting Items for Systematic Reviews and Meta-Analyses) guidelines[63]. Searches were performed in Web of Science (WoS) (Core collection) and Scopus between July and November 2020. The search was limited to studies published in English between January 1990 and November 2020. The first search used the following terms: (ectotherm* OR insect*) AND (thermal OR heat OR cold OR chill OR temperature) AND (min* OR max* OR critical OR surviv* OR lethal) AND (plastic* OR (phenotyp* plastic*)) OR acclim* OR stress OR tolerance) NOT (plant* OR tree* OR fung* OR mammal* OR marsup* OR bird* OR reptile* OR lizard* OR snake* amphib* OR frog* OR toad* OR fish* OR newt*).

As WoS only had three hits before 1990, articles were only included from the period of 1990 to 2020 to reduce bias between the two databases. Coverage of the literature was assessed using previously mentioned meta-analyses which examined acclimation in ectotherms. Of the four articles (post 1990) on insects included in an analysis by Seebacher et al. (2015), all were found in the present literature search[22]. Twenty-one articles (post 1990) on insect species were included in Gunderson et al.'s 2015 study, nine of which were picked up by the first search[16]. Therefore coverage of the literature was deemed to be insufficient, so an additional search was completed between October and November 2020 using more comprehensive search terms. The following were used: (insect* OR ecopteran* OR archaeognatha OR bristletail* OR ecoptera* OR ecopteran* OR *lice OR *louse OR Psocopter* OR blattodea* OR cockroach* OR ecopter* OR ecoptera* OR ecoptera* OR dermaptera* OR earwig* OR orthoptera* OR grasshopper* OR cricket* OR ecoptera* OR mantis* OR mantid* OR ephemeroptera* OR ecopt* OR ecopteran* OR phasmid* OR ecopter* OR ecopteran* OR isoptera* OR termite* OR ecopteran* OR thrip* OR hemiptera* OR *bug* OR cicada* OR aphid OR *hopper* OR ecopteran* OR webspinner* OR web-spinner* OR zoraptera* OR endopterygot* OR megaloptera* OR hymenoptera* OR wasp* OR ants OR ant OR bee OR bees OR coleoptera* OR beetle* OR lepidoptera* OR ecoptera* OR moth* OR caterpillar* OR ecopteran* OR ecoptera* OR ecopteran* OR flea* OR diptera* OR *fly OR *flies OR mosquito* OR ecopteran* OR lacewing* OR antlion* OR ecopteran* OR raphidioptera* OR strepsiptera*) AND (thermal OR heat OR cold OR chill OR temperature) AND (min* OR max* OR critical OR surviv* OR lethal) AND (plastic* OR (phenotyp* plastic*)) OR acclim* OR stress OR tolerance). These search terms resulted in full coverage of the literature from the aforementioned meta-analyses.

### Eligibility criteria

The exclusion procedure is summarised in Supplementary Fig. 1. In total, the two databases found 12,139 unique results. Study abstracts were scanned manually for suitability by a single observer (HW) and selected studies were further examined by their methodology. Studies were selected for data extraction if they described dynamic tolerance assays where $CT_{max}$ or $CT_{min}$ was measured by ramping the temperature until a specified endpoint. We chose to only evaluate dynamic studies as it was a common metric used in thermal tolerance assays, removed additional sources of methodological heterogeneity, and was a metric already synthesised in other meta-analyses. We required studies to have at least two temperature treatments (including studies where a single treatment was compared to a control), perform a temperature acclimation treatment (all durations of acclimation, including acute hardening and longer-term chronic acclimation, were included and fluctuating temperatures were allowed), and be undertaken in a laboratory. Studies were not included if any variables in addition to temperature were modified (excluding named moderators). Studies were also excluded if the endpoint was recorded for a proportion of the insects assayed only (e.g., $CT_{max}80$).

## Data extraction and effect size calculation

Data were extracted (arithmetic mean, standard deviation (SD), sample size (N)) from 60 and 52 articles, comprising 92 and 74 species, for $CT_{max}$ and $CT_{min}$ respectively, from tables or text directly, from Supplementary Information, or directly requested from the authors when not available. The full dataset can be found in Supplementary Data 1 and a full reference list of all papers included in the meta-analysis can be found in Supplementary Data 2. Four studies (cited in Supplementary Data 3) where a very large number of insects were measured were removed from the $CT_{max}$ dataset as the unusually large sample sizes ($n > 700$) grossly inflated the study weight and it was deemed that this number of insects could not be accurately assessed to the individual level in one run. When only presented in graphical form, data were digitised from Figures using R package 'metaDigitise' (Version 1.0.1). Axes were calibrated using the longest distance possible to increase accuracy. If the error bars were obscured by the data points, the full size of the data point was taken as the error as a conservative measure. If not directly stated, sample sizes were calculated from degrees of freedom, and where the resulting numbers were non-integer, the sample size was rounded down. Where a range of sample sizes were stated, the smallest was always taken.

Acclimation Response Ratio (ARR) was calculated for $CT_{max}$ and $CT_{min}$ from the raw data using $ARR = \frac{CT_{[T_2]} - CT_{[T_1]}}{T_2 - T_1}$, where CTL is the critical thermal limit ($CT_{max}$ or $CT_{min}$) and T is the acclimation temperature[45]. This results in a positive ARR if heat acclimation increases $CT_{max}$ or if cold acclimation decreases $CT_{min}$. The standardized slope can be interpreted as a change in CTL for each degree change in acclimation temperature. As in Pottier et al. (2021), when more than two acclimation temperatures were reported, pairwise comparisons were made (e.g., 10–12 °C, 12–15 °C, 15–20 °C). We calculate multiple ARR measures rather than deriving a single slope per study in order to capture potential (and likely) non-linearity in the relationship between acclimation temperature and CTL. This meant that some responses were used in ARR calculations twice. To account for this, a variance covariance (VCV) correlation matrix was used to reduce the weight of dependent observations (see 'Statistical analysis'). The variance was calculated as: $Var = \left(\frac{1}{T_2 - T_1}\right)^2 \left(\frac{SD^2_{[T1]}}{N_{[T1]}} + \frac{SD^2_{[T2]}}{N_{[T2]}}\right)$, where SD is the standard deviation and N is the adjusted sample size.

## Moderator variables

Prior to the analysis, predictions were made regarding the chosen moderators and submitted to Turnitin. Moderators were extracted either from the study itself or from published studies and databases. We included all durations of acclimation, resulting in 19% of effect sizes ($k = 265$) with acclimation treatments under 24 hours, 35% ($k = 486$) between 1 and 7 days, and 45% ($k = 623$) for over a week. As some studies stated the duration of acclimation treatment in life stages rather than a metric of time, 35% ($k = 478$) of data were missing. Unfortunately, this meant data available were biased to shorter acclimation times as longer acclimations were usually stated in life stages. The stage at which the insect was acclimated was during the juvenile stage for 23% of effect sizes ($k = 310$), adults for 51% ($k = 694$), several life stages for 24% ($k = 334$), and several generations for 3% ($k = 36$). We also recorded whether the acclimation treatment and assay were within the same life stage ($k = 1004$) or over different life stages ($k = 370$), so that we could test for preliminary evidence of the effect of the temperature-size rule on ARR (Supplementary Table 3 and 4). For mass data, 10% ($k = 132$) came directly from the paper, 69% ($k = 946$) from the wider literature, with the remaining 21% missing ($k = 296$). If wet (fresh) body mass was not stated, data were first obtained from studies for the same species within our database, otherwise, we searched the wider literature. References for studies from which mass estimates were extracted can be found in

Supplementary Data 4. Where only dry mass was available, estimates of wet mass were made by using water balance estimates of closely related species found in Hadley (1994)[64]. For latitude, where only a place name was given, we chose a midpoint within this area and used Google Maps to drop a marker in the middle of the location specified. Some studies did not provide detail of the source of a laboratory population, meaning that 14.6% ($k = 202$) of data were missing for latitude.

## Statistical analysis

All analyses were completed in R version 4.0.3. The following sources of non-independence were identified and considered: phylogenetic relationships, non-phylogenetic species-related effects (e.g., shared ecology), population effects (e.g., same collection site), study effects (ARRs calculated from the same study), pairwise comparisons for ARR calculations, and within study effects (effect size ID; variability in the true effects within studies). Phylogenetic trees were constructed in the Open Tree of Life and R packages 'rotl' (Version 3.0.11) and 'ape' (Version 5.5) (for full trees, see Supplementary Fig. 2)[65]. A phylogenetic correlation matrix was constructed based on hypothetical relatedness of species. A VCV matrix was constructed to account for dependant observations due to pairwise comparisons during ARR calculation. Branch lengths were assigned following Grafen's method. The VCV matrix did not explain any of the variation in the data, so was excluded from subsequent models. The final random effect structure was study ID, phylogeny, species ID, and effect size ID. Although the random effect structure was not the best fit for $CT_{min}$ data (study ID and effect size ID only), for ease of interpretation we ran models with the same structure.

The R package 'metafor' (Version 3.0-2) was used to perform multi-level, random effects models[66]. All models were run with the chosen random effect structure, with data for $CT_{max}$ and $CT_{min}$ run separately. Intercept models were fitted to assess the overall effect of acclimation on $CT_{max}$ and $CT_{min}$. We tested moderators by running models with each moderator individually. We then fitted full models with all moderators and used the 'Dredge' function from the MuMIn package (Version 1.43.17) to assess which combination had the best fit ($\Delta AICc \leq 2$)[67]. The multivariate models excluded latitude, mass and acclimation duration as these moderators did not have a complete dataset available, which would have reduced power and may have affected the results. As differences between variables were small, we used conditional averages rather than full averages which tend to be more conservative and bias small results towards zero[67]. Full-average model statistics can be found in the Supplementary Tables 8 and 10. Statistical significance was assumed when 95% confidence intervals (95% CIs) did not span zero or, when comparing groups, 95% CIs did not overlap. Residuals were assessed for homogeneity of variance between groups visually. Where residuals were heterogeneous the robust.rma.mv function from the metafor package was used. $I^2$ was calculated for each model to assess heterogeneity (proportion of variance not attributed to sampling error). We calculated the overall amount of heterogeneity, $I^2$ total, as well as the heterogeneity explained by each of the random effects.

## Sensitivity analyses and publication bias

Leave-one-out analyses were performed by iteratively removing one family, species, or order to determine if any influential groups affected the model outcome. Analyses were also completed without Drosophilidae, and with Drosophilidae only, and without data where the acclimation treatment was a fluctuating temperature, results for which are reported in the Supplementary Tables 13–17. Publication bias was assessed by funnel plot and Egger's regression test (Supplementary Tables 18–23). Egger's regression test was performed by fitting standard error as a unique moderator and as part of the best fit model[68]. As publication bias was identified, we then fitted Standard Error$^2$ as a

moderator to predict mean ARRs corrected for publication bias. A model was also fitted with study year as a moderator to examine temporal biases (Supplementary Tables 24-25). All code was adapted from Pottier et al. (2021)[36] and Macartney et al. (2019)[69]. Figures were constructed using the orchaRd package (Version 0.0.0.9) and by adapting code from Pottier et al. (2021)[36].

## Reporting summary
Further information on research design is available in the Nature Research Reporting Summary linked to this article.

## Data availability
The raw and processed data used in this study are available in the Supplementary Files (Supplementary Data 1) and on the OSF database under accession code: https://osf.io/cbhv4/.

## Code availability
The code used to analyse data is available on the OSF database: https://osf.io/cbhv4/.

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

## Acknowledgements

We would like to acknowledge the authors who contributed research articles for data extraction, references for which can be found in Supplementary Data 2. Financial support was provided by a Royal Society Dorothy Hodgkin Fellowship (DH140236) and BBSRC grant (BB/P006159/1) to SE, BBSRC studentship to H.W., and a University of Bristol GCRF pump-priming grant to S.E. and J.T. P.P. was supported by a UNSW Scientia Doctoral Scholarship.

## Author contributions

H.W. completed the data collection and analysis. P.P. contributed code and input to the analysis. S.E. and J.T. jointly supervised the work. H.W. led writing of the manuscript and all authors contributed to the development of drafts, interpretation of results, revisions, and responses to referees.

## Competing interests

The authors declare no competing interests.
