## [Peer Review File · Nature Communications]

Reviewer comments, first round review -

Reviewer #1 (Remarks to the Author):

This paper presents a well designed, considered and executed meta-analysis on the thermal tolerance of a wide range of insect species under acclimation, in order to test a range of hypotheses related to their ability to adapt and survive temperature events (and including climate change).

The work is worthy of publication in a prestigious journal. However, there are some significant issue that I feel warrants more discussion and analysis than currently presented.

One near universal response of ectotherms to warming during ontogeny is a shift in body size at stage (the temperature-size rule). This phenotypic plastic response is believed to be an adaptive response that in part aids the organism to deal with warming. As such, short acclimations (or of single life stages) do not consider such a change, and this might have important implications on the findings, in comparison to the situation when organisms experience warming over multiple generations. While there is a ontogenetic and life cycle data include in the plots and data collection, little is made of this. Interestingly orthopterans are one of the orders that almost always increase in size with warming, whilst most other taxa tend to decline in adult size with warming. I note the outlying nature of orthopterans in your analysis too (albeit from unfortunately a tiny data set). See: Horne et al (2015) Temperature-size responses match latitudinal-size clines in arthropods, revealing critical differences between aquatic and terrestrial species. *Ecology Letters* 18: 327-335. doi: 10.1111/ele.124; and Forster et al (2012) Warming-induced reductions in body size are greater in aquatic than terrestrial species. *Proceedings of the National Academy of Sciences USA* 109: 19310-19314. doi: 10.1073/pnas.1210460109

I note you found juveniles to be more plastic in their thermal tolerance than adults, yet changes in size at maturity with warming may act to offset some of this difference between life stages, as body size changes develop over ontogeny and are commonly strongest in adults.

The authors need to address the issues I have highlighted much more explicitly in terms of the acclimation period and in terms of their prediction/hypotheses (including the ontogeny section in Box 1). The temperature-size rule is near ubiquitous and has too important a potential in relation to temperature tolerance by life stage, acclimation period and ontogeny to have been completely ignored in your work.

I would therefore like a more detailed analysis and account of the acclimation period with response to the ontogeny and generation / across generations.

Reviewer #2 (Remarks to the Author):

The paper "Plasticity of thermal limits is unlikely to rescue insects from rapid climate change" by Weaving et al. presents a comprehensive analysis of literature data investigating the magnitude of acclimation responses for upper and lower critical limits in insects. The overall finding from this study concludes that the magnitude of the acclimation response on critical thermal limits is far from compensatory. Thus, critical limits for heat CT_{max} only increase approx. 0.1 C per degree of acclimation and the acclimation response for cold is approx. 0.15 C per degree of acclimation. It is therefore argued that the limited acclimation response is insufficient to safeguard insects from the anticipated climate change.

The interest in this question is not completely new and a number of recent papers (most of which are referenced in this submission) have also addressed this question. Below are 6 examples of other reviews discussing plasticity of insects in the context of climate change (Papers in bold are cited here already) – Most of these reviews discussing this problem reaching a similar conclusion (that plasticity is moderate and insufficient to buffer climate change) and at least the last two

studies mentioned below have similar comparative analytical approach analysing patterns of ARR (Acclimation response ratio) as is presented here. (Note however that the study submitted here is undertaking a MUCH more comprehensive analysis of this issue)

1. Sgro et al., (2016) What Can Plasticity Contribute to Insect Responses to Climate Change? *Annu. Rev. Entomol.*
2. van Heerwaarden et al., (2016) Limited scope for plasticity to increase upper thermal limits. *Funct. Ecol.*
3. van Heerwaarden, and Kellermann, (2020). Does Plasticity Trade Off With Basal Heat Tolerance? *TREE*
4. Kellermann and Van Heerwaarden (2019). Terrestrial insects and climate change: adaptive responses in key traits. *Physiol. Entomol.*
5. Gunderson and Stillman (2015). Plasticity in thermal tolerance has limited potential to buffer ectotherms from global warming. *Proc R. Soc Biol*
6. Sørensen et al. (2016). Evolutionary and ecological patterns of thermal acclimation capacity in *Drosophila*: is it important for keeping up with climate change? *Cur. Op. Ins. Sci.*

Even though the potency of the insect acclimatory response has been discussed and investigated previously, there is no doubt that the submitted study sets a new (and much higher) standard. The submitted study is much more comprehensive and thorough. Most importantly the meta-study submitted here covers a much broader list of insect orders (often grouped according to hemi- and holometabolic orders), the study also considers developmental stage, type of knockdown assay and even provides a good argument for a slight publication bias (which might overestimate thermal plasticity). The methods and statistical analysis is presented convincingly and carefully and I find all figures to be very well organised presenting both the overall results and details with great care. Following the comprehensive analysis it is found that these other sources of variance (developmental stage, assay type and order) do contribute a little to patterns found. However, the main conclusions reached here are generally very well aligned with the conclusions of the studies mentioned above, both in terms of magnitude of the ARR (around 0.1 to 0.2 C/C), and also in terms of the broader inference in relation to climate change.

Reading through the MS I made a number of small comments which are listed below.

Page 1, line 1. Title - I am not a big fan of the title (although it is catchy). I have never really the idea that plasticity could rescue any ectotherm from climate change. If we imagine that thermal tolerance is restricting distribution, then we must also assume that the species currently living at the (warmest) edge of the distribution are already exposed to the warmest tolerable temperatures that have therefore also already activated phenotypic plasticity. Plasticity is not an unused buffer that is activated upon climate change. In my view the question of plasticity is therefore mainly relevant if climate change imposes higher degrees of thermal variance (and there are some arguments for that).

Page 2, line 36. I don't think it is entirely unknown if there is potential for this rescue. As mentioned in my first point, I don't personally follow the idea, but even if I accept the idea of such rescue, the 6 papers mentioned (and some not mentioned here), have already reached the conclusion that plasticity is insufficient. Also in this line - there is a reference to "Animals", consider to use insects instead.

Page 2, line 37. I am not a great fan of splitting the data up data in effects sizes (there may be some good arguments statistically, but from my understanding of the results, this approach just introduces a lot of variance that is tied to study ID). If I understand the approach correctly, then the analysis here gets 5 effect sizes from a study that has investigated CT_{max} in a species acclimated to 6 temperatures (effect size 1 is from temp 1-2, effect size 2 is from 2-3 and so on). I think it would be much more fair just to use all data and make one slope. I doubt that the authors will follow this suggestion (and there may be some arguments for that), but I think it would be fair to present already in the abstract how many species are included for heat and cold respectfully.

Page 2, line 40: publication bias could cause a slight overestimation (consider to emphasise that this is a small effect - although the analysis of this is new)

Page 2, line 43-45: as mentioned already a few times, I struggle with the idea that plasticity can rescue anything (that it is not already "rescuing").

Page 3, Line 53: consider to cite Sunday (she has a number of papers that show reasonably good association of CTmax with distribution in insects (for example Sunday et al., Thermal-safety margins and the necessity of thermoregulatory behavior across latitude and elevation), there are also several papers showing very good correlation between insect cold tolerance and distribution (particularly when appropriate thermal tolerance assays are used)

Page 3, Line 59-61: This is a little confusing to me. Browsing through your list of references, the vast majority of studies use insects reared at different (benign) temperatures, and the differences in CTmax are not likely to be linked to hsp. What you are describing here is a heat hardening response, which is rather acute, and probably physiologically very different from a chronic acclimation response. I don't think the hsp example serves well. Modification in membranes would maybe be a better example. Consider if/when you want acclimation vs hardening examples.

Page 3, line 69: I don't think the Seebacher reference has CTL data (but it is a good study 9)

Page 3, Line 72: both Gunderson/Stillman and Sørensen et al also have CTmin.

Page 3, line 76: I don't think it is correct that this is the first formal study of ARR (see for example Sørensen et al and Gunderson and Stillman), in fact I think it would be generally fair if the introduction was better at introducing that this question has been asked (and answered) previously. You could summarise what is known prior to this analysis and then argue how your analysis is in much greater depth, with a larger dataset and more comprehensive analysis. I find the present version a little too dismissive of previous discussions and analysis (particularly considering that the main conclusions are largely the same as here).

Page 4, line 88: I don't think there is a formal analysis of the relevance to climate change, and I think actually it is unclear what we can say about these plasticity patterns in relation to that (this will depend on changes in temperature variability),

Page 5, line 148: recent studies by Jørgensen provide a strong mathematical argument for this (Jørgensen et al. (2021) A unifying model to estimate thermal tolerance limits in ectotherms across static, dynamic and fluctuating exposures to thermal stress, scientific reports), but the situation is probably very different for cold where RCH occurs during gradual cooling (Kelty and Lee (1999). Induction of rapid cold hardening by cooling at ecologically relevant rates in *Drosophila melanogaster*. JEB).

Page 5, line 153: I don't understand the argument why there is an assumption that some endpoints should respond differently – explain the underlying idea better.

Page 5, line 156: It is my experience from working with insects that constant temperatures is the odd treatment. Insects exposed to moderate variability do better in a lot of tests, and my immediate suggestion would therefore be that insects exposed to natural variance would already have activated their "plastic capacity", and thus be able to respond less. In other words I would think there is much more scope for plasticity in the animal from the constant environment. (just like there is much larger scope for athletic improvement in the untrained person than there is in the already well trained person). This is also what you find (page 11, line 322)

Page 6. I think you could be more clear on this page with how many species that are included in the analysis. Also, I still think it would be fine just to use one slope when many acclimation treatments were included in one study.

Page 7-11: very thorough and well-explained analysis.

Page 12 – general. As commented on in the introduction there could be made more effort in referring that the findings have many similarities to previous studies/reviews (some of these

previous studies also analysed associations to latitude, CTmin vs CTmax etc.) and overall the main effects reported here are very consistent with previous studies on insects (and I am sure there is also some literature on this in other ectotherms).

Page 13, line 392: Is it really essential to account for diverse aspects of methodology? Many of the new aspects investigated here are clearly relevant, but as presented clearly, they contribute very little (even if significant) to the patterns of variance in plasticity.

Page 13, line 399 and onwards: As I recall the Klok reference does not investigate different mechanisms of CTmin or max (but rather it investigates the same putative mechanism – oxygen limitation – and finds little support). I miss a little better discussion of differences and similarities. Both heat and cold knockdown (coma) are very likely to be caused by CNS failure (Spreading depression). The difference is that heat death occurs at almost the same temperature, while an insect can be in coma for considerable time without causing injury. There are some general reviews on both heat and cold tolerance that have discussed the physiology of these phenotypes (although it is largely still unclear what causes heat death in insects)

Page 13, line 409: I struggle a little to understand the argument regarding concrete ceilings, and probably most of all I struggle to find out how this discussion is relevant to the present paper.

Page 13, line 413: I agree that it is contentious if climate extremes correlate well with distribution. Some of the Sunday studies indicate that this is the case in general, and several drosophila studies indicate that this is the case within a species group where CTLs are measured consistently. BUT obviously CTLs are not the only trait explaining distribution of species (but they do explain a considerable part of the variation (particularly cold limits)). Maybe present references for and against?

Page 14, line 426: maybe provide a stronger argument for why a phylogenetic signal results in limited adaptation. Not sure I understand the causality.

Paper in general: Since I have mainly commented on small issues that I think could be strengthened slightly, I would like to end by saying, that the paper was generally very well written and easy to follow, the figures are really good. I enjoyed reading the paper and I believe this study sets a new and better standard for the discussion on plasticity and climate change in insects.

Reviewer #3 (Remarks to the Author):

Weaving and colleagues present an interesting metanalysis of thermal plasticity in insects framed in the context of extreme events precipitated by climate change and the broad role of insects in agriculture and human health. The work appears sound overall and will be of value to the ecological physiology field. The ms is well written and I appreciated the detailed documentation of analyses that was included in the supp documents. In my opinion, the novelty and impact of the findings may, however, be somewhat overstated. The contextualisation of the research could also be much better developed if it is intended to support broader statements around how insects could be affected by extreme climate events.

The overarching claims of this ms, perhaps best summarised in the final sentence of the abstract (“current experimental studies show plasticity of thermal limits is unlikely to rescue most insect species from extreme climatic events”), are a bit of a leap from the results presented. At no point have the authors established how much plasticity is required to ‘rescue’ insect species. Is anything near to a 1:1 ARR needed to ‘rescue’ species? What is known about insect thermal tolerances relative to current climate extremes (e.g. the vast literature on warming tolerances and thermal safety margins)? Without additional analyses (or at least more in depth discussion of the literature around CTLs relative to environmental extremes), claims about ‘rescue’ (or not) should be tempered. The finding of juveniles being more plastic than adults is interesting, though by the authors own admission this finding was not robust to the ‘knock out’ approach used to minimise the effects of taxonomic bias in the analyses.

Authors' claims of novelty (e.g. "Here, we undertake the first formal, systematic meta-analysis of experimental studies on the plasticity") are perhaps a bit overzealous. I was surprised to see little discussion Barley et al's recently published metanalysis of plasticity in thermal traits (doi.org/10.1098/rspb.2021.0765), which directly examined the same 'plasticity ~ climate variability' relationship that Weaving et al. have looked at here. Barley et al. also looked at trade offs with basal thermal tolerances, an interaction that I suggest should also be considered in this metanalysis (Barley et al found trade-offs to be much more important than lat/seasonality trends).

SPECIFIC COMMENTS

L070 – has such publication bias already been established or is you testing/finding of this bias novel?

L083 – as mentioned above I don't think these results help identify which insect groups are most vulnerable to climate change – basal tolerance and future env would need to be incorporated into analyses to do that

L100 – did you look at absolute latitude? How did you account for differences in seasonality between N and S (e.g. S Hemisphere generally less variable at the same latitude). Why not use a direct measure of seasonality like diff between min and max temp at sampling location?

L104 – can it be assumed that fw aquatic environments are more stable than terrestrial? I've predominantly seen this applied in marine vs terrestrial comparisons.

L111 – if smaller insects have reduced ability to exploit microclimates then shouldn't they be more plastic to compensate for this? (i.e. they can't move as readily so will have evolved a physiol solution)

L155 – is there an interaction between body size and sex given sexual dimorphism in many insect spp?

L150 – in my experience it is rare to see a lethal measure classified as a CT (usually it would be LT100 or something similar). It's certainly an interesting question to examine how the differing plasticity of CTs and LTs but I think in order to do that properly, studies that defined lethal temps as LTs should also be added to the metanalysis.

L152 – suggest rephrasing this as an explicit hypothesis

L167 – please state the exact cut off date – were papers published in December 2020 were excluded as the search ended in Nov?

L190 – suggest stating that these slopes are being used as the effect size for subsequent analyses

L199 – excellent that a priori hypotheses were filed somewhere! – is there a doi associated with this so readers can confirm date and original hypotheses?

L211 – suggest stating how branch lengths were estimated and what that could mean for phylo analyses – from reading the supp info it looks like Grafen's method was used.

L245 – thank you for the detailed walkthrough of analyses presented in your supp info!

L259 – is k the number of species or number of studies? Orthoptera is listed as k = 4 here and k = 1 elsewhere in the ms.

L298 – as above, suggest adding LT studies to the metanalyses to make this comparison, otherwise it will only capture a handful of studies that have classified lethal temps as CTs.

L337/338 – results are duplicated between parentheses, no p values for significance testing

L356 – not sure anyone expects/hypothesises 'complete compensation' from acclimation treatments, nor is it essential for 'rescue'

L360 – the role/interaction of plasticity in relation to current or future environmental extreme was not examined

L363 – migration, behaviour, and their interaction with physiol would benefit from much more detailed discussion (+ reference to broader literature)

L363-382 – removal of Orthoptera eliminated the difference observed (L375). Thus, the findings discussed in this paragraph are not robust to the authors' own bias checking/correction methods. Suggest the certainty of this par and its meaning/impact be toned down as a consequence.

L391 – suggest expanding on why lab-reared and field fresh differed contrary to expectations. Any data on the number of generations for lab-reared populations? FF vs F2 vs F50+ (for some dros lines).

L396-410 – interesting finding and very well articulated

[redacted]

Response to Reviewer Comments -

We have carefully considered

each of the main comments and addressed the following as suggested:

- 1) The data we have available in our synthesis would not allow for the type of analyses Reviewer 1 suggests as body size (mass) was largely taken from the wider literature, not the specific study, and there were no cases where mass was taken at the juvenile and adult stage, where the temperature-size rule would become apparent. However, we have given a better account of body size, ontogeny and acclimation duration in the methodology as Reviewer 1 suggests, making the scope of our dataset clearer. We have also provided an extra piece of analysis to show that there is no difference in acclimation response ratio (ARR) between insects where the acclimation treatment and the critical thermal limit (CTL) assay were in different life stages (where the temperature size rule would be apparent), and those where the treatment and assay were in the same life stage (no effect of the temperature-size rule). We have also taken care to explain that the temperature-size rule may interact with some of the plasticity predictions in Box 1. (See response to point R1.2).*
- 2) Reviewer 3 suggested that lethal temperature assays should be added to the study. As explained in further detail in response to R3.11, we do not include this in the present study for three main reasons. First, our aim is to investigate the extent and variation of plasticity of insect thermal limits, focusing on plasticity in critical thermal minima and maxima. We did include some measures where CTLs were defined as death, as one of our questions was to investigate how ARR's differ between endpoint definitions. We appreciate the Reviewer's point that expanding the dataset to include lethal temperature would allow a comprehensive comparison of how plasticity varies between lethal and critical thermal limits, but this is a different question. Second (and relatedly), lethal temperature assays have a very different methodology to critical thermal limit (CTL) assays: they are typically static thermal assays, rather than temperature ramps, and are given at the temperature where a percentage of a population die, rather than CTLs which are taken at the individual level. It would be possible to convert these data into comparable effect sizes to ARR's. However, we would need to start the meta-analysis over from the beginning with a new list of search terms, literature search, paper selection, data extraction and analysis. This would take at least another year of work and was not our original aim.*

Finally, we note that a recent meta-analysis on developmental plasticity in ectotherms by one of manuscript authors finds no difference in the plasticity of CTmax and LT50 (preprint <https://www.authorea.com/doi/full/10.22541/au.164573834.49438532/v1>). This suggests that the expansion of the dataset to include lethal temperature assays may not change our general conclusions about the extent of thermal limit plasticity in insects.

- 3) *We appreciate Reviewer 2's concern over multiple effect sizes and we had considered deriving slopes from each study to use as single effect sizes. However, these would not capture potentially important non-linear relationships between CTLs and acclimation temperature that are often observed (with time, e.g. Weldon et al. 2011 JTB, and with acclimation temperature e.g. Terblanche et al. 2006 Am J Trop Med Hyg). We therefore decided to use multiple comparisons to capture these non-linear relationships and formally accounted for the use of multiple estimates per study in the statistical models (as explained in response to R2.4)*

See below for a detailed response to each reviewer's comments and the changes made. Line numbers refer to changes in the finalised manuscript file, rather than the track changes file.

Reviewer #1

R1.1

This paper presents a well designed, considered and executed meta-analysis on the thermal tolerance of a wide range of insect species under acclimation, in order to test a range of hypotheses related to their ability to adapt and survive temperature events (and including climate change).

The work is worthy of publication in a prestigious journal. However, there are some significant issue that I feel warrants more discussion and analysis than currently presented.

>>>RESPONSE: *We appreciate this generally positive appraisal of our work. We have explained below how we have incorporated more discussion on this important issue, and why we are unable to include the suggested analysis. We have completed a piece of extra analysis (explained in detail below) to show the temperature-size rule (TSR) does not have a significant effect on ARR overall, although we admit that due to interacting effects (and the lack of study-specific body size data), this only provides preliminary evidence.*

R1.2

One near universal response of ectotherms to warming during ontogeny is a shift in body size at stage (the temperature-size rule). This phenotypic plastic response is believed to be an adaptive response that in part aids the organism to deal with warming. As such, short acclimations (or of single life stages) do not consider such a change, and this might have important implications on the findings, in comparison to the situation when organisms experience warming over multiple generations. While there is a ontogenetic and life cycle data include in the plots and data collection, little is made of this. Interestingly orthopterans are one of the orders that almost always increase in size with warming, whilst most other taxa tend to decline in adult size with warming. I note the outlying nature of orthopterans in your analysis too (albeit from unfortunately a tiny data set). See: Horne et al (2015) Temperature-size responses match latitudinal-size clines in arthropods, revealing critical differences between aquatic and terrestrial species. *Ecology Letters* 18: 327-335. doi: 10.1111/ele.124; and Forster et al (2012) Warming-induced reductions in body size are greater in aquatic than terrestrial species. *Proceedings of the National Academy of Sciences USA* 109: 19310-19314. doi: 10.1073/pnas.1210460109

I note you found juveniles to be more plastic in their thermal tolerance than adults, yet changes in size at maturity with warming may act to offset some of this difference between life stages, as body size changes develop over ontogeny and are commonly strongest in adults.

The authors need to address the issues I have highlighted much more explicitly in terms of the acclimation period and in terms of their prediction/hypotheses (including the ontogeny section in Box 1). The temperature-size rule is near ubiquitous and has too important a potential in relation to temperature tolerance by life stage, acclimation period and ontogeny to have been completely ignored in your work.

I would therefore like a more detailed analysis and account of the acclimation period with response to the ontogeny and generation / across generations.

>>>RESPONSE: *The focus of our study was the extent to which acclimation allows for shifts in thermal tolerance across insects. We appreciate, however, that the temperature-size rule (TSR) could play an important role in thermal plasticity and our original manuscript did not acknowledge this sufficiently.*

That said, we do not have the data to address this question specifically (although we report some preliminary analysis below). Most of our measures on body mass were extracted from

the wider literature simply to indicate a species' relative size. Only a limited number of studies (132/1374 effect sizes) provided mass data for the same individuals used in the acclimation treatment, and of these, none of them report mass before and after the thermal treatment, which is essential to investigate the TSR and any potential influence that it might have on plasticity of CTLs. We now explain the origin of the body mass data more clearly in the methodology (l. 411-412), and we also provide more detail on the length of acclimation period in the study and at what stage the acclimation treatment took place (l. 401-412). "We included all lengths of acclimation, resulting in 19% of effect sizes ($k = 265$) with acclimation treatments under 24 hours, 35% ($k = 486$) between 1 and 7 days, and 45% ($k = 623$) for over a week. As some studies stated the duration of acclimation treatment in life stages rather than a metric of time, 35% ($k = 478$) of data were missing. ... The stage at which the insect was acclimated was at the juvenile stage in 23% of effect sizes ($k = 310$), adults in 51% ($k = 694$), several life stages in 24% ($k = 334$), and several generations in 3% ($k = 36$). We also recorded whether the acclimation treatment and assay were within the same life stage ($k = 1004$) or over different life stages ($k = 370$) so that we could test for preliminary evidence of the temperature-size rule." This will enable the reader to see more clearly what can and cannot be answered using these data.

We have now conducted additional analysis comparing ARR between groups where acclimation and the CTL assay occurred within a life stage (and would not be affected by TSR) or across several life stages (and thus would be potentially affected by TSR). For both CT_{max} (CT_{max} ARR_{within-between} = 0.001; 95% CI = -0.030, 0.032) and CT_{min} (CT_{min} ARR_{within-between} = 0.038; 95% CI = -0.037, 0.113), we found no significant difference between groups. We have included these results in Supplementary Tables 3 and 4. This provides some evidence that the TSR may not have a large effect on ARR. However, relationships between size, ontogeny and acclimation length are likely to be much more complicated. In the wider literature, there is some evidence to support this – a study looking at TSR and plasticity of thermal tolerance in acorn ants found size did not change under 5 different rearing temperatures, and so heat and cold tolerance plasticity were independent of size (Yilmaz et al. *J Therm Biol* 2019). Other broader interspecific studies, including our analysis, have found no relationship between size and thermal tolerance or its plasticity (e.g. Scharf et al. *Evol Biol* 2015; Janion-Scheepers et al 2018 PNAS).

We agree that TSR is important conceptually and we now remind readers of the TSR where relevant (e.g. in box 1 predictions; l.122-124): "As an added complication, insects under high developmental temperatures generally become smaller (the temperature-size rule (TSR)³²), which may act counter to our predictions."

At the same time, we've been clearer that we cannot undertake a formal test of this TSR-CTL plasticity interaction here (l. 124-125) "A formal test of TSR could not be undertaken in the present study as most mass estimates were derived from the wider literature (see Methodology for specific details)". It would however make for interesting future work and

we have made this comment in the discussion (l. 314 – 317). "Our preliminary analyses investigating the temperature-size rule (Supplementary Table 3 and 4) found no difference between groups where acclimation treatment and CTL assay were within a life stage, or over different life stages. However, it would be interesting to investigate this further where study-specific mass/size data are available."

Reviewer #2 (Remarks to the Author):

R2.1

The paper "Plasticity of thermal limits is unlikely to rescue insects from rapid climate change" by Weaving et al. presents a comprehensive analysis of literature data investigating the magnitude of acclimation responses for upper and lower critical limits in insects. The overall finding from this study concludes that the magnitude of the acclimation response on critical thermal limits is far from compensatory. Thus, critical limits for heat CT_{max} only increase approx. 0.1 C per degree of acclimation and the acclimation response for cold is approx. 0.15 C per degree of acclimation. It is therefore argued that this limited acclimation response is insufficient to safeguard insects from the anticipated climate change.

The interest in this question is not completely new and a number of recent papers (most of which are referenced in this submission) have also addressed this question. Below are 6 examples of other reviews discussing plasticity of insects in the context of climate change (Papers in bold are cited here already) – Most of these reviews discussing this problem reaching a similar conclusion (that plasticity is moderate and insufficient to buffer climate change) and at least the last two studies mentioned below have similar comparative analytical approach analysing patterns of ARR (Acclimation response ratio) as is presented here. (Note however that the study submitted here is undertaking a MUCH more comprehensive analysis of this issue)

- 1. Sgro et al., (2016) What Can Plasticity Contribute to Insect Responses to Climate Change? Annu. Rev. Entomol.**
- 2. van Heerwaarden et al., (2016) Limited scope for plasticity to increase upper thermal limits. Funct. Ecol.**
- 3. van Heerwaarden, and Kellermann, (2020). Does Plasticity Trade Off With Basal Heat Tolerance? TREE**
- 4. Kellermann and Van Heerwaarden (2019). Terrestrial insects and climate change: adaptive responses in key traits. Physiol. Entomol.**
- 5. Gunderson and Stillman (2015). Plasticity in thermal tolerance has limited potential to buffer ectotherms from global warming. Proc R. Soc Biol**
- 6. Sørensen et al. (2016). Evolutionary and ecological patterns of thermal acclimation capacity in *Drosophila*: is it important for keeping up with climate change? Cur. Op. Ins. Sci.**

Even though the potency of the insect acclamatory response has been discussed and investigated previously, there is no doubt that the submitted study sets a new (and much higher) standard. The submitted study is much more comprehensive and thorough. Most importantly the meta-study submitted here covers a much broader list of insect orders (often grouped according to hemi- and holometabolic orders), the study also considers developmental stage, type of knockdown assay and even provides a good argument for a slight publication bias (which might overestimate thermal plasticity). The methods and statistical analysis is presented convincingly and carefully and I find all figures to be very well organised presenting both the overall results and details with great care. Following the comprehensive analysis it is found that these other sources of variance (developmental stage, assay type and order) do contribute a little to patterns found. However, the main conclusions reached here are generally very well aligned with the conclusions of the studies mentioned above, both in terms of magnitude of the ARR (around 0.1 to 0.2 C/C), and also in terms of the broader inference in relation to climate change.

>>>**RESPONSE:** *We are delighted that the reviewer sees the value in our study, being considerably more comprehensive and detailed than previous work. We have rewritten part of the introduction to frame our paper building upon others rather than emphasising its novelty (l. 64-69).*

"Recent systematic reviews and formal meta-analyses across ectotherms have assessed plasticity of CTLs and described broad-scale patterns of variation in plasticity¹⁶⁻²¹. Generally, these studies find weak plasticity of CTLs, concluding that this plasticity has limited potential to aid survival of ectothermic species from climate change."

It builds confidence in the results of the work that these findings have been found on ectotherms generally in previous studies. We have also added the Sorensen et al. 2016 and Kellerman & Heerwaarden, 2019 reference.

R2.2

Reading through the MS I made a number of small comments which are listed below.

Page 1, line 1. Title - I am not a big fan of the title (although it is catchy). I have never really the idea that plasticity could rescue any ectotherm from climate change. If we imagine that thermal tolerance is restricting distribution, then we must also assume that the species currently living at the (warmest) edge of the distribution are already exposed to the warmest tolerable temperatures that have therefore also already activated phenotypic plasticity. Plasticity is not an unused buffer that is activated upon climate change. In my view the question of plasticity

is therefore mainly relevant if climate change imposes higher degrees of thermal variance (and there are some arguments for that).

>>>RESPONSE: *Good point. We have changed the title to: "Meta-analysis reveals weak but pervasive plasticity in insect thermal limits". We agree that plasticity is relevant to increasing thermal variance, as addressed in the first paragraph of the introduction (l. 50-52). We have also moved away from the idea of "rescuing" insects as, in agreement with Reviewer 1, we also did not expect that plasticity would fully compensate for the effects of climate change. These changes can be seen in the title and first paragraph of the discussion (l. 1; 265-270).*

R2.3

Page 2, line 36. I don't think it is entirely unknown if there is potential for this rescue. As mentioned in my first point, I don't personally follow the idea, but even if I accept the idea of such rescue, the 6 papers mentioned (and some not mentioned here), have already reached the conclusion that plasticity is insufficient. Also in this line – there is a reference to "Animals", consider to use insects instead.

>>>RESPONSE: *As stated above (response to points R2.1 and R2.2), we have moved away from the idea of "rescue" throughout the manuscript and we have reframed the introduction substantially.*

We have changed 'animals' to 'insects' in the abstract (l. 34).

R2.4

Page 2, line 37. I am not a great fan of splitting the data up data in effects sizes (there may be some good arguments statistically, but from my understanding of the results, this approach just introduces a lot of variance that is tied to study ID). If I understand the approach correctly, then the analysis here gets 5 effect sizes from a study that has investigated CTmax in a species acclimated to 6 temperatures (effects size on is from temp 1-2, effect size 2 is from 2-3 and so on). I think it would be much more fair just to use all data and make one slope. I doubt that the authors will follow this suggestion (and there may be some arguments for that), but I think it would be fair to present already in the abstract how many species are included for heat and cold respectfully.

>>>RESPONSE: *Our approach is indeed understood correctly by the reviewer. We appreciate the argument on using slopes (and had considered analysing the data as such), however there are two main reasons we use multiple ARR measures instead. First, and most importantly, reducing the data from multiple acclimation temperatures to one slope fails to capture the non-linear relationships between acclimation temperature and CTmax which has been shown in several studies (see Editor response, point 1). Having one slope*

essentially assumes that this relationship is always linear and constant across acclimation temperatures, which is not always true. In fact, it seems like there is a concrete ceiling that constrain CTmax and its plasticity, so the relationship might be more asymptotic than linear. Including multiple ARR measures better accounts for such non-linear relationships between acclimation temperature and CTmax. We now explain this in the Methodology at l. 391-393. Moreover, the fact that multiple estimates per study (as well as multiple comparisons involving the same data) were used was formally accounted for in our statistical models (as explained in lines 424-425). Second, such an approach has been used in previous studies (e.g., Morley et al. 2019; Gunderson & Stillman 2015), so it is informative to use similar indices while addressing similar questions.

As requested, we have presented the number of species in the abstract (l. 36). We have not separated out the number for CTmax and CTmin due to word count limitations, but this information can be found in the methodology (l. 377-378).

R2.5

Page 2, line 40: publication bias could cause a slight overestimation (consider to emphasise that this is a small effect – although the analysis of this is new)

>>>RESPONSE: Agreed, there was only a very small effect of publication bias and we now state 'small but significant publication bias' to account for this (l.39-40).

R2.6

Page 2, line 43-45: as mentioned already a few times, I struggle with the idea that plasticity can rescue anything (that it is not already “rescuing”).

>>>RESPONSE: We have generally toned down the idea of a “rescue” effect, as outlined in the comments above.

R2.7

Page 3, Line 53: consider to cite Sunday (she has a number of papers that show reasonably good association of CTmax with distribution in insects (for example Sunday et al., Thermal-safety margins and the necessity of thermoregulatory behavior across latitude and elevation), there are also several papers showing very good correlation between insect cold tolerance and distribution (particularly when appropriate thermal tolerance assays are used)

>>>RESPONSE: We have read and cited the Sunday et al 2014 paper – thank you. As this paper covers both cold and heat tolerance, we have not added any further papers due to the Nature Communications limit on references.

R2.8

Page 3, Line 59-61: This is a little confusing to me. Browsing through your list of references, the vast majority of studies use insects reared at different (benign)

temperatures, and the differences in CTmax are not likely to be linked to hsp. What you are describing here is a heat hardening response, which is rather acute, and probably physiologically very different from a chronic acclimation response. I don't think the hsp example serves well. Modification in membranes would maybe be a better example. Consider if/when you want acclimation vs hardening examples.

>>>**RESPONSE:** We considered both acute hardening (hours) and chronic acclimation (days-weeks) in the meta-analysis, so we consider the example of heat shock proteins still a relevant mechanism. We have now, however, added in another example, of changes to the phospholipid membrane (l. 57-58). We have also made it clearer in the methodology that all lengths of acclimation time were considered (l. 401-403). There were fewer hardening examples (19% (265/1374)) than acclimation examples but both were considered.

R2.9

Page 3, line 69: I don't think the Seebacher reference has CTL data (but it is a good study)

>>>**RESPONSE:** Agreed, Seebacher et al. 2015 contains acclimation data for "metabolic rates, heart rates, enzyme activities and locomotor performance" so we have removed the reference here – thank you!

R2.10

Page 3, Line 72: both Gunderson/stillman and Sørensen et al also have CTmin.

>>>**RESPONSE:** Both the cited studies do include both CTmax and CTmin. However, Sorensen is not a meta-analysis so it's difficult to make comparisons. The others cited (Barley et al. Proc. R. Soc. B Biol. Sci. 2021, Rohr et al Ecol. Lett. 2018, Seebacher et al Nat. Clim. Chang. 2015) do not consider lower temperatures, so we still think it's a valid point that CTmin is less investigated than CTmax. However, due to the restructuring of the introduction, we have removed this sentence anyway.

R2.11

Page 3, line 76: I don't think it is correct that this is the first formal study of ARR (see for example Sørensen et al and Gundersen and Stillman), in fact I think it would be generally fair if the introduction was better at introducing that this question has been asked (and answered) previously. You could summarise what is known prior to this analysis and then argue how your analysis is in much greater depth, with a larger dataset and more comprehensive analysis. I find the present version a little to dismissive of previous discussions and analysis (particularly considering that the main conclusions are largely the same as here).

>>>**RESPONSE:** We meant that this was the first study looking at insects exclusively and in depth with a host of novel moderators. However, we agree that this sentence can be easily misconstrued so have removed the word 'novelty'. We have also generally changed

the emphasis towards the rationale of a study of greater depth with a focus on insects (l. 64-69; 73-78) (see also responses to point R2.1). We also try to be clear when a particular outcome or result is confirmatory or perhaps more novel to better distinguish what has been found previously (e.g. l. 82-83; 265-268).

R2.12

Page 4, line 88: I don't think there is a formal analysis of the relevance to climate change, and I think actually it is unclear what we can say about these plasticity patterns in relation to that (this will depend on changes in temperature variability),

>>>RESPONSE: *We have removed 'and their relevance to climate change'.*

R2.13

Page 5, line 148: recent studies by Jørgensen provide a strong mathematical argument for this (Jørgensen et al. (2021) A unifying model to estimate thermal tolerance limits in ectotherms across static, dynamic and fluctuating exposures to thermal stress, scientific reports), but the situation is probably very different for cold where RCH occurs during gradual cooling (Kelty and Lee (1999). Induction of rapid cold hardening by cooling at ecologically relevant rates in *Drosophila melanogaster*. JEB).

>>>RESPONSE: *Really interesting paper – thank you! We have added this reference to Box 1 (l.143).*

R2.14

Page 5, line 153: I don't understand the argument why there is an assumption that some endpoints should respond differently – explain the underlying idea better.

>>>RESPONSE: *We have altered Box 1 to make our point clearer (l. 151-154). We predicted that behavioural responses e.g. response to prodding will be more plastic than if the endpoint is measured as death.*

R2.15

Page 5, line 156: It is my experience from working with insects that constant temperatures is the odd treatment. Insects exposed to moderate variability do better in a lot of tests, and my imideate suggestion would therefore be that insects exposed to natural variance would already have activated their "plastic capacity", and thus be able to respond less. In other words I would think there is much more scope for plasiticity in the animal from the constant environment. (just like there is much larger scope for athletic improvement in the untrained person than there is in the already well trained person). This is also what you find (page 11, line 322)

>>>RESPONSE: *This is an interesting argument. Our thoughts were that after many generations in the constant temperature of the laboratory, potentially costly plasticity*

would be lost as it has been shown that there is a genetic component to plasticity (e.g. Oostra et al. 2018 Nat. Comms.). In contrast, in the field, insects exposed to a greater temperature variation need to be more plastic.

We added an argument to the discussion for why we may have found laboratory insects to have higher ARR for CTmin (l. 308-311). We believe that this could be due to more factors influencing and interacting with tolerance in the field-collected individuals. In the lab more factors are controlled (e.g. diet, age, thermal history all affect estimates) and thus the signal may be clearer in lab-reared specimens.

R2.16

Page 6. I think you could be more clear on this page with how many species that are included in the analysis. Also, I still think it would be fine just to use one slope when many acclimation treatments were included in one study.

>>>**RESPONSE:** We have made it clearer how many species were included in the analysis by adding how many species were included over both measures of CTmax and CTmin (l. 377-378). We have given our rationale above for including several ARRs per study rather than one slope (response to R2.4).

R2.17

Page 7-11: very thorough and well-explained analysis.

>>>**RESPONSE:** Thank you!

R2.18

Page 12 – general. As commented on in the introduction there could be made more effort in referring that the findings have many similarities to previous studies/reviews (some of these previous studies also analysed associations to latitude, CTmin vs CTmax etc.) and overall the main effects reported here are very consistent with previous studies on insects (and I am sure there is also some literature on this in other ectotherms).

>>>**RESPONSE:** We reframed the introduction by outlining results from similar studies, and how this work builds upon them (l. 64-69; 82-83). Additionally, we have added detail to the discussion, comparing our results to similar studies (l. 265-268; 350-351). "*Indeed, in Gunderson and Stillman's 2015¹⁶ analysis, insects had the weakest responses of all ectothermic groups and Morley et al.'s 2019²⁰ analysis illustrated a similar pattern (when excluding high latitude species).*"

R2.19

Page 13, line 392: Is it really essential to account for diverse aspects of methodology? Many of the new aspects investigated here are clearly relevant, but

as presented clearly, they contribute very little (even if significant) to the patterns of variance in plasticity.

>>>RESPONSE: *True, we have removed 'essential' and rephrased (l. 317-318). "Our findings indicate the need to consider diverse aspects of methodology and population history in future comparative analyses."*

R2.20

Page 13, line 399 and onwards: As I recall the Klok reference does not investigate different mechanisms of CT_{min} or max (but rather it investigates the same putative mechanism – oxygen limitation – and finds little support). I miss a little better discussion of differences and similarities. Both heat and cold knockdown (coma) are very likely to be caused by CNS failure (Spreading depression). The difference is that heat death occurs at almost the same temperature, while an insect can be in coma for considerable time without causing injury. There are some general reviews on both heat and cold tolerance that have discussed the physiology of these phenotypes (although it is largely still unclear what causes heat death in insects)

>>>RESPONSE: *We have removed the Klok reference and altered this paragraph using reviews to form our discussion, moving away from spreading depolarisation (l. 321-328). "CT_{max} is often lethal, occurring at the same temperature as heat death. There are relatively few studies on the mechanisms of heat death, but the breakdown of membrane function may cause the impairment of ion pumps, nutrient transport and mitochondrial function, leading to the loss of metabolic control and homeostasis, and, finally, cell death⁵⁵. In contrast, CT_{min} usually results in a non-lethal chill coma, where an insect may recover fully. The mechanisms of chill coma are also poorly understood, but are likely driven by the breakdown of ionic homeostasis due to the effect of low temperature on ATPases, ion channels and the lipid membrane⁵⁶."*

R2.21

Page 13, line 409: I struggle a little to understand the argument regarding concrete ceilings, and probably most of all I struggle to find out how this discussion is relevant to the present paper.

>>>RESPONSE: *We have rewritten some of this paragraph to make the meaning and relevance of the concrete ceilings argument clearer (l.329-334). The argument is really about mechanistic constraints or a lack thereof that may occur in upper limits; it is a concept that appears in the literature quite frequently (especially in fish physiology) so we feel it should be addressed. "We also detected a phylogenetic signal in CT_{max} in our models, which was not observed for CT_{min}. This may reflect evolutionary constraints for CT_{max}, such as high fitness costs or substantial genetic changes required to modify upper thermal limits, causing related species to share similar thermal responses⁵⁷. If upper thermal limits*

cannot evolve easily due to these constraints, an organism's current thermal limits will dictate the kind of environments in which it can survive. These differences create a 'concrete ceiling' for CT_{max} , where physiological barriers prevent extensive evolution and perhaps also restrict plasticity⁵⁸." Moreover, Reviewer 3 was positive about this paragraph and its relevance (point R3.26)

R2.22

Page 13, line 413: I agree that it is contentious if climate extremes correlate well with distribution. Some of the Sunday studies indicate that this is the case in general, and several drosophila studies indicate that this is the case within a species group where CTLs are measured consistently. BUT obviously CTLs are not the only trait explaining distribution of species (but they do explain a considerable part of the variation (particularly cold limits)). Maybe present references for and against?

>>>RESPONSE: We have reused a reference from Sunday to show that there is both evidence for and against. "However, there is some evidence that CTLs do not correlate well with species abundance or distribution, and therefore might not be the best predictors for assessing climate change impacts (but see³)^{59,60}."

R2.23

Page 14, line 426: maybe provide a stronger argument for why a phylogenetic signal results in limited adaptation. Not sure I understand the causality.

>>>RESPONSE: We have added some detail to help explain this better (l.329-334). See comment R2.21.

R2.24

Paper in general: Since I have mainly commented on small issues that I think could be strengthened slightly, I would like to end by saying, that the paper was generally very well written and easy to follow, the figures are really good. I enjoyed reading the paper and I believe this study sets a new and better standard for the discussion on plasticity and climate change in insects.

>>>RESPONSE: We appreciate your positive and constructive feedback – thank you!

Reviewer 3

R3.1

Weaving and colleagues present an interesting metanalysis of thermal plasticity in insects framed in the context of extreme events precipitated by climate change

and the broad role of insects in agriculture and human health. The work appears sound overall and will be of value to the ecological physiology field. The ms is well written and I appreciated the detailed documentation of analyses that was included in the supp documents. In my opinion, the novelty and impact of the findings may, however, be somewhat overstated. The contextualisation of the research could also be much better developed if it is intended to support broader statements around how insects could be affected by extreme climate events.

>>>**RESPONSE:** *We have toned down mention of novelty throughout the manuscript and framed the research as more detailed and specific to insects with unique moderators than previous studies. We have also changed the introduction to improve the contextualisation (l. 64-69; 82-83, see responses R2.11, R2.18 to reviewer 2 above).*

R3.2

The overarching claims of this ms, perhaps best summarised in the final sentence of the abstract (“current experimental studies show plasticity of thermal limits is unlikely to rescue most insect species from extreme climatic events”), are a bit of a leap from the results presented. At no point have the authors established how much plasticity is required to ‘rescue’ insect species. Is anything near to a 1:1 ARR needed to ‘rescue’ species? What is known about insect thermal tolerances relative to current climate extremes (e.g. the vast literature on warming tolerances and thermal safety margins)? Without additional analyses (or at least more in depth discussion of the literature around CTLs relative to environmental extremes), claims about ‘rescue’ (or not) should be tempered. The finding of juveniles being more plastic than adults is interesting, though by the authors own admission this finding was not robust to the ‘knock out’ approach used to minimise the effects of taxonomic bias in the analyses.

>>>**RESPONSE:** *Reviewer 2 also questioned the idea of plasticity ‘recuing’ insects from climate change (point R2.2, R2.3, R2.6): we agree this is unclear and we have now removed such phrasing from the manuscript. We have also added detail on thermal safety margins to the discussion (l. 268-270).*

As for the finding that juveniles were more plastic than adults, this result was robust to knockout. It was the finding that hemimetabolous insects were more plastic than holometabolous insects that was not. We have split the discussion about development stage and development type (hemimetabolous or holometabolous) into two paragraphs to make this clearer.

R3.4

Authors’ claims of novelty (e.g. “Here, we undertake the first formal, systematic meta-analysis of experimental studies on the plasticity”) are perhaps a bit

overzealous. I was surprised to see little discussion Barley et al's recently published meta-analysis of plasticity in thermal traits (ProcB doi.org/10.1098/rspb.2021.0765), which directly examined the same 'plasticity ~ climate variability' relationship that Weaving et al. have looked at here. Barley et al. also looked at trade offs with basal thermal tolerances, an interaction that I suggest should also be considered in this meta-analysis (Barley et al found trade-offs to be much more important than lat/seasonality trends).

>>>RESPONSE: *As stated earlier, we have moved away from novelty, framing the research as unique due to its moderators and depth.*

We did read (and cite) the Barley paper. There is currently debate over the methodology of comparing basal tolerance with plasticity thereof in this way (see van Heerwaarden, and Kellermann, (2020). Does Plasticity Trade Off With Basal Heat Tolerance? TREE). In short, the regression involves comparing heat tolerance and plasticity, but these are not statistically independent as heat tolerance is used on both axes. Therefore, they are almost necessarily negatively correlated because of the regression to the mean. For this reason, we did not investigate this trend in our meta-analysis and so felt that a discussion on the basal tolerance vs plasticity argument was not directly relevant to the current paper.

R3.5

L070 – has such publication bias already been established or is you testing/finding of this bias novel?

>>>RESPONSE: *Good question. Seebacher et al. 2015, Rohr et al 2018, Gunderson and Stillman 2015 and Gunderson and Stillman 2017 make no reference to publication bias and did not produce funnel plots.*

Barley et al 2021 did calculate a measure of publication bias (Rosenberg's 'fail-safe number'), finding no evidence of publication bias. However, using Egger's regression test with multi-level meta-regressions is a much more powerful means to establish publication bias (as demonstrated in the current meta-analysis). We also included far more studies, Barley's estimate being based on 20 studies in total.

That said, the line the reviewer refers to is not about publication bias but bias in the methodology of other meta-analyses which examine plasticity of ARR. Several select the response of highest effect rather than extracting data from all treatments as we have done in the present study. We have removed this argument from the manuscript in the general reframing of the introduction.

R3.6

L083 – as mentioned above I don't think these results help identify which insect groups are most vulnerable to climate change – basal tolerance and future env would need to be incorporated into analyses to do that

>>>**RESPONSE:** *This is a fair point and we have now removed this statement.*

R3.7

L100 – did you look at absolute latitude? How did you account for differences in seasonality between N and S (e.g. S Hemisphere generally less variable at the same latitude). Why not use a direct measure of seasonality like diff between min and max temp at sampling location?

>>>**RESPONSE:** *Yes, we looked at absolute latitude. We did not account for differences in seasonality between North and South so have rerun the analysis putting the data into bins of 10 degrees latitude. We still found no effect of latitude on ARR (see full effect sizes for latitude analyses in the Supplementary Tables 3 and 4). Sampling location was not always reported, particularly with laboratory strains of insects, so a more considered approach will be needed for determining seasonality effects, perhaps on a subset of the data, which was beyond the scope of this study.*

R3.8

L104 – can it be assumed that fw aquatic environments are more stable than terrestrial? I've predominantly seen this applied in marine vs terrestrial comparisons.

>>>**RESPONSE:** *Yes, I think we can assume this as water is a poor conductor of heat so even in freshwater water temperature will be less variable than air temperature. Although, it is true that changes in body temperature are faster in water than air and that there are generally less opportunities to thermoregulate in water. So for this reason we may have seen the opposite effect.*

R3.9

L111 – if smaller insects have reduced ability to exploit microclimates then shouldn't they be more plastic to compensate for this? (i.e. they can't move as readily so will have evolved a physiol solution)

>>>**RESPONSE:** *Yes, this argument should have been part of the reason why we might **not** find plasticity increases with size. Thank you for spotting this, we have altered the section (l. 110-112).*

R3.10

L155 – is there an interaction between body size and sex given sexual dimorphism in many insect spp?

>>>**RESPONSE:** *Unfortunately, we do not have the data for this question. Only two of the studies provided sex-specific mass data and only 132/1374 effect sizes were study specific, with most mass values used are species-level averages from the wider literature. We have*

made it clearer where we obtained our mass estimates in the methodology so the reader is clear what can be achieved with our data (l. 411-412).

R3.11

L150 – in my experience it is rare to see a lethal measure classified as a CT (usually it would be LT100 or something similar). It's certainly an interesting question to examine how the differing plasticity of CTs and LTs but I think in order to do that properly, studies that defined lethal temps as LTs should also be added to the metanalysis.

>>>RESPONSE: We agree with Reviewer #3's point that it would be interesting to examine LTs as well as CTLs. However, we would argue that this is a standalone paper from the current work and would require the same amount of time as the present paper (1 year) to redo the literature search, screening, data extraction and analysis. Studies measuring LTs often use a very different methodology for measuring thermal tolerance – often the thermal stress is prolonged (e.g. 24h) and the survival of a proportion of individuals are measured. Estimates (e.g. LT50) are then obtained from multiple cohorts of animals with regression-based approaches. Therefore, the measure of dispersion is always standard error (estimated from model estimates of the logistic or probit regression) and the formulas used to calculate the sampling variance of ARR need to be modified accordingly. One of our authors, PP, has used LTs in a previous meta-analysis (Pottier et al. <http://doi.org/10.22541/au.164573834.49438532/v1>) and found that standard errors are often not reported so there is also often a need to obtain raw survival data (further lengthening the process) and regressions need to be completed. Moreover, they did not find evidence for an influence of metric type (i.e. CTmax or LT50) on thermal plasticity estimates.

In the present dataset, we only included studies which stated that they were measuring CTmax or CTmin. Two of these studies we included measured CTmin or CTmax at the point of death (a total of 7 effect sizes). An option is to remove these from the study, however one of the aims of the paper was to look at differences in the end point at which the critical temperature was measured, and if this affected ARR.

R3.12

L152 – suggest rephrasing this as an explicit hypothesis

>>>RESPONSE: We have rephrased as an explicit hypothesis (l. 152-154).

R3.13

L167 – please state the exact cut off date – were papers published in December 2020 were excluded as the search ended in Nov?

>>>RESPONSE: Yes, this was the case. We have made it clearer the dates in which papers were included (l. 361-362).

R3.14

L190 – suggest stating that these slopes are being used as the effect size for subsequent analyses

>>>**RESPONSE:** *Correct, we have stated this (l. 391-392).*

R3.15

L199 – excellent that a priori hypotheses were filed somewhere! – is there a doi associated with this so readers can confirm date and original hypotheses?

>>>**RESPONSE:** *Unfortunately, we do not have a doi for readers (we will be sure to pre-register our hypotheses in future). However, it is possible to verify these are indeed a priori as they are reported in an internal university report which has been submitted (with date stamp) to Turnitin.*

R3.16

L211 – suggest stating how branch lengths were estimated and what that could mean for phylo analyses – from reading the supp info it looks like Grafen’s method was used.

>>>**RESPONSE:** *We have now clarified that Grafen’s method was used (l. 424).*

R3.17

L245 – thank you for the detailed walkthrough of analyses presented in your supp info!

>>>**RESPONSE:** *We are delighted the reviewer appreciates this detail.*

R3.18

L259 – is k the number of species or number of studies? Orthoptera is listed as k = 4 here and k = 1 elsewhere in the ms.

>>>**RESPONSE:** *K is the number of individual effect sizes, and some studies may report multiple effect sizes. Orthoptera is k = 3 for CTmin, and k = 1 for CTmax. When we refer to k = 4, this is in reference to the total effect sizes. When we discuss removal of Orthoptera from the CTmax dataset, this involves excluding k = 1.*

R3.19

L298 – as above, suggest adding LT studies to the metaanalyses to make this comparison, otherwise it will only capture a handful of studies that have classified lethal temps as CTs.

>>>**RESPONSE:** *Please see the arguments made above (point R3.11) for why this is an interesting suggestion but out of scope of the current study.*

R3.20

L337/338 – results are duplicated between parentheses, no p values for significance testing

>>>RESPONSE: *The results in parentheses are not duplicated, rather we provide the same statistical outcome (rounded) results for both CTmin and CTmax models. We have not used p values because we already presented 95% confidence intervals, which can be used to assess statistical significance and present less concerns than p values (Benjamin et al. 2017 Nat. Hum.).*

R3.21

L356 – not sure anyone expects/hypothesises ‘complete compensation’ from acclimation treatments, nor is it essential for ‘rescue’

>>>RESPONSE: *Similar comment to reviewer #2 (point R2.2, R2.3, R2.6, R3.2). We have removed the idea of rescue/complete compensation.*

R3.22

L360 – the role/interaction of plasticity in relation to current or future environmental extreme was not examined

>>>RESPONSE: *Fair point and we have removed this statement.*

R3.23

L363 – migration, behaviour, and their interaction with physiol would benefit from much more detailed discussion (+ reference to broader literature)

>>>RESPONSE: *We have now included more discussion on behavioural thermoregulation and migration of insect species (l. 273-276).*

R3.24

L363-382 – removal of Orthoptera eliminated the difference observed (L375). Thus, the findings discussed in this paragraph are not robust to the authors’ own bias checking/correction methods. Suggest the certainty of this par and its meaning/impact be toned down as a consequence.

>>>RESPONSE: *The result that juveniles are more plastic than adults was robust to removing Orthoptera, so we do not see the need to tone down this part of the paragraph. We have split these results into two paragraphs to ensure there is not confusion between the results.*

With regards the contrast between hemi- and holometabolous insects (which was eliminated with removal of Orthoptera), we still think the result that hemimetabolous insect are more plastic is worthy of discussion as all four of the models investigating this found that hemimetabolous insects were more plastic (although not significant).

Nevertheless, we have now restructured the paragraph to make it clearer that the result is not robust (l. 293-303).

R3.25

L391 – suggest expanding on why lab-reared and field fresh differed contrary to expectations. Any data on the number of generations for lab-reared populations? FF vs F2 vs F50+ (for some dros lines).

>>>RESPONSE: We have expanded on why laboratory populations may be have greater responses (l. 308-311).

We did manage to collect some data on the number of generations. However, often the authors provided this information in numbers of years (44% of effect sizes), rather than number of generations (45% of effect sizes, with over half of these being 0 values for field-caught individuals). As we could not find an adequate lifespan database to calculate number of generations from years, we decided to keep this as lab vs field.

Unfortunately, Drosophilidae are just as susceptible to this problem – only 35% of data (208/584) were reported in number of generations in the laboratory.

R3.26

L396-410 – interesting finding and very well articulated

>>>RESPONSE: Thank you.

Reviewer comments, second round review -

Reviewer #2 (Remarks to the Author):

To the Authors

I find that the revision of this manuscript is a considerable upgrade of an already fine manuscript. All the points I raised in the initial revision have been handled carefully and I therefore only have two minor specific points and a more general suggestion that mainly represents personal preference.

Page 5 line 143: Reference 44 (Jørgensen et al) is used to argue that different acclimation lead to different degrees of stress. This is not the point of that paper (unless you specifically discuss hardening which occurs at stressful temperatures). The Jørgensen paper clearly introduces a temperature below which there is no net accumulation of heat injury. As an example a beetle will not compile injure from acclimation at either 10 or 20C if injury does not start to accumulate before temperature is above 25C and the argument presented in line 143 is therefore misleading. I do indeed believe the paper is quite relevant for the question here, but probably much more in the section about methodology (Lines 135-139)

Page 9 line 269: As I understand the Sunday paper many ectotherms (many insects included) do NOT experience AIR temperature above CTmax, but due to radiation from the sun, the operative BODY temperature will approach CTmax. I don't think this is clear in this sentence that reference to the air temperature.

My general comment is that I find that the introduction and box is very well references and points to many relevant papers related to the analysis presented here. In the discussion part of the paper, many of the same references could also be relevant to the discussions and the authors could consider to reference the literature a little more in this section where they relate their meta analysis to the hypothesis presented. This is simply my personal preference and I am not suggesting that the discussion is rewritten, but rather that the authors consider if some of the literature introduced initially is also relevant to mention in the discussion (especially because the referencing style is not disturbing to the reader).

Reviewer #3 (Remarks to the Author):

Weaving and colleagues have done an excellent job revising their ms and responding to all three sets of reviewer comments - I look forward to seeing this work published.

Two minor suggested edits:

In the Abstract the authors state "Insects can enhance their critical thermal limits through acclimation,..." the way this is phrased I interpret acclimation as something that insects are actively seeking out to 'supercharge' themselves - obviously not the case or the intended meaning. I recommend switching the subject from 'insects' to 'thermal limits' - i.e. "Insect critical thermal limits can be enhanced through acclimation". Apologies for only mentioning this now, I had it flagged in

my notes on R0 but didn't transcribe it into my initial review.

L067 - "...mechanism has limited potential to aid survival of ectothermic species from climate change" - odd wording, survival 'under' climate change?

REVIEWERS' COMMENTS

Reviewer #2 (Remarks to the Author):

To the Authors

I find that the revision of this manuscript is a considerable upgrade of an already fine manuscript. All the points I raised in the initial revision have been handled carefully and I therefore only have two minor specific points and a more general suggestion that mainly represents personal preference.

>>>RESPONSE:

Thank you, we appreciated your helpful comments!

Page 5 line 143: Reference 44 (Jørgensen et al) is used to argue that different acclimation lead to different degrees of stress. This is not the point of that paper (unless you specifically discuss hardening which occurs at stressful temperatures). The Jørgensen paper clearly introduces a temperature below which there is no net accumulation of heat injury. As an example a beetle will not compile injure from acclimation at either 10 or 20C if injury does not start to accumulate before temperature is above 25C and the argument presented in line 143 is therefore misleading. I do indeed believe the paper is quite relevant for the question here, but probably much more in the section about methodology (Lines 135-139)

>>>RESPONSE:

Thank you for pointing this out. We have now given two examples where injury accumulation or the 'detrimental acclimation hypothesis' is discussed (Loeschcke & Hoffmann, Trends Ecol. Evol. 2002; Cossins & Bowler, 1987), adding these references instead of the Jørgensen reference.

Lines 150-151 "As acclimation can be stressful, we expect smaller ARR for longer acclimation times as injury accumulates exponentially with time (e.g. discussed in ^{44,45})".

Page 9 line 269: As I understand the Sunday paper many ectotherms (many insects included) do NOT experience AIR temperature above CTmax, but due to radiation from the sun, the operative BODY temperature will approach CTmax. I don't think this is clear in this sentence that reference to the air temperature.

>>>RESPONSE:

We have changed the sentence to reflect that Sunday was discussing body temperatures rather than air temperatures (and added that the paper was referring to ectotherms in exposed environments).

Lines 280-283 "Under our current climate, some evidence suggests that the majority of ectothermic species are close to or without a thermal safety margin, as operative body temperatures in exposed environments often match or exceed physiological limits (albeit requiring a number of simplifying assumptions)³."

My general comment is that I find that the introduction and box is very well references and points to many relevant papers related to the analysis presented here. In the discussion part of the paper, many of the same references could also be relevant to the discussions and the authors could consider to reference the literature a little more in this section where they relate their meta analysis to the hypothesis presented. This is simply my personal preference and I am not suggesting that the discussion is rewritten, but rather that the authors consider if some of the literature introduced

initially is also relevant to mention in the discussion (especially because the referencing style is not disturbing to the reader).

>>>RESPONSE:

We have added several references from the introduction and a priori predictions section to the discussion to give better depth (Lines 291, 300). We have also added a little further detail in methodology section of the discussion to relate better to the predictions.

Lines 326-330 "Notably, no relationship was found between ARR and acclimation duration or ramp rate, perhaps owing to complex interactions which were not investigated in this analysis, such as those between ramp rate and acclimation, nutrition and body condition, and interval time between the acclimation treatment and endpoint^{53,54}."

Reviewer #3 (Remarks to the Author):

Weaving and colleagues have done an excellent job revising their ms and responding to all three sets of reviewer comments - I look forward to seeing this work published.

>>>RESPONSE:

Thank you for your constructive comments which helped greatly improve the manuscript!

Two minor suggested edits:

In the Abstract the authors state "Insects can enhance their critical thermal limits through acclimation,..." the way this is phrased I interpret acclimation as something that insects are actively seeking out to 'supercharge' themselves - obviously not the case or the intended meaning. I recommend switching the subject from 'insects' to 'thermal limits' - i.e. "Insect critical thermal limits can be enhanced through acclimation". Apologies for only mentioning this now, I had it flagged in my notes on R0 but didn't transcribe it into my initial review.

>>>RESPONSE:

Agreed. We have changed this sentence to that described above (Lines 34-35).

L067 - "...mechanism has limited potential to aid survival of ectothermic species from climate change" - odd wording, survival 'under' climate change?

>>>RESPONSE:

We have altered this sentence to reflect the reviewer's comment by switching 'from' to 'under' (Line 67).